# Sublethal systemic LPS in mice enables gut-luminal pathogens to bloom through oxygen species-mediated microbiota inhibition

Sanne Kroon [1], Dejan Malcic[1], Lena Weidert[1,2], Lea Bircher[2],
Leonardo Boldt [3,4,5], Philipp Christen [1], Patrick Kiefer[1], Anna Sintsova[1],
Bidong D. Nguyen[1], Manja Barthel[1], Yves Steiger [1], Melanie Clerc [1],
Mathias K.-M. Herzog[1], Carmen Chen[6], Ersin Gül[1], Benoit Guery[6], Emma Slack [2],
Shinichi Sunagawa [1], Julia A. Vorholt [1], Lisa Maier [3,4,5],
Christophe Lacroix [2], Annika Hausmann [1,2,7] ✉ & Wolf-Dietrich Hardt [1] ✉

Endotoxin-driven systemic immune activation is a common hallmark across various clinical conditions. During acute critical illness, elevated plasma lipopolysaccharide triggers non-specific systemic immune activation. In addition, a compositional shift in the gut microbiota, including an increase in gut-luminal opportunistic pathogens, is observed. Whether a causal link exists between acute endotoxemia and abundance of gut-luminal opportunistic pathogens is incompletely understood. Here, we model acute, pathophysiological lipopolysaccharide concentrations in mice and show that systemic exposure promotes a 100–10'000-fold expansion of *Klebsiella pneumoniae*, *Escherichia coli*, *Enterococcus faecium* and *Salmonella* Typhimurium in the gut within one day, without overt enteropathy. Mechanistically, this is driven by a Toll-like receptor 4-dependent increase in gut-luminal oxygen species levels, which transiently halts microbiota fermentation and fuels growth of gut-luminal facultative anaerobic pathogens through oxidative respiration. Thus, systemic immune activation transiently perturbs microbiota homeostasis and favours opportunistic pathogens, potentially increasing the risk of infection in critically ill patients.

The healthy gut is colonised by a commensal microbiota that releases large amounts of lipopolysaccharide (LPS), a potent immune stimulant, into the gut lumen (~80 μg LPS ml⁻¹ in mouse caecum)[1]. The healthy gut epithelial barrier is impermeable to gut-luminal LPS, resulting in low levels of LPS in human plasma (≤0.2 ng ml⁻¹)[2–5]. In the acute phase of critical illness, LPS originating from the gut lumen or extraintestinal sites can reach a concentration of 2–10 ng ml⁻¹ in the plasma, commonly referred to as endotoxemia[2,6,7]. Endotoxemia typically triggers acute, non-specific systemic immune activation[3]. In addition, critically ill patients frequently present with shifts in gut microbiota composition, including increased abundance of opportunistic pathogens like *Enterococcus* spp. or *Enterobacteriaceae*, which is

[1]Institute of Microbiology, Department of Biology, ETH Zürich, Zürich, Switzerland. [2]Institute of Food, Nutrition and Health, Department of Health Sciences and Technology, ETH Zürich, Zürich, Switzerland. [3]Interfaculty Institute of Microbiology and Infection Medicine Tübingen, University of Tübingen, Tübingen, Germany. [4]M3 Research Center for Malignome, Metabolome and Microbiome, University Hospital Tübingen, Tübingen, Germany. [5]Cluster of Excellence 'Controlling Microbes to Fight Infections', University of Tübingen, Tübingen, Germany. [6]Infectious Diseases Service, Lausanne University Hospital and University of Lausanne, Lausanne, Switzerland. [7]reNEW - Novo Nordisk Foundation Center for Stem Cell Medicine, University of Copenhagen, Copenhagen, Denmark. ✉e-mail: annika.hausmann@hest.ethz.ch; hardt@micro.biol.ethz.ch

associated with severe clinical outcomes[8–11]. The risk of microbiota shifts in these patients is often exacerbated by clinical interventions, such as the use of antibiotics, which are common during an extended hospitalisation. However, these late-stage interventions alone do not fully explain the early onset of gut-luminal pathogen blooms observed in these patients[12,13]. We hypothesise that acute endotoxemia represents an early factor eliciting such blooms. Whether there is a mechanistic link between acute endotoxin-driven systemic immune activation and gut-luminal opportunistic pathogen blooms remains incompletely understood.

*Salmonella* Typhimurium (*S*. Tm) is a prevalent pathogen causing self-limiting gastroenteritis in most healthy individuals while posing life-threatening risks for immunocompromised individuals, children, and elderly. *S*. Tm employs virulence factors for gut tissue invasion, replication, and systemic dissemination, eliciting both innate and adaptive immune responses[14,15]. The gut microbiota plays a key role in protecting against enteropathogens like *S*. Tm, but it also serves as a reservoir for opportunistic pathogens, such as *Escherichia coli*, *Klebsiella pneumoniae*, and *Enterococcus faecium*[16,17]. These opportunistic pathogens typically colonise the intestine of healthy individuals asymptomatically, but can result in systemic infection upon host perturbation as observed in critically ill patients[18–21]. Moreover, as members of the *Enterobacteriaceae* family, *E. coli* and *K. pneumoniae* can elicit systemic immune responses similar to those against *S*. Tm upon systemic dissemination[18–21]. These observations emphasise the delicate balance between commensal and pathogenic states of bacteria.

The unperturbed microbiota provides colonisation resistance against pathogens by competition for nutrients and secretion of inhibitory metabolites[22–26]. Microbiota fermentation of dietary substrates, such as complex carbohydrates and amino acids, produces short-chain fatty acids (SCFA) that not only serve as an energy source for epithelial cells but also inhibit pathogens[27–30]. Conversely, colonisation resistance is impaired upon microbiota perturbation by intestinal inflammation or antibiotic treatment, suppressing fermentation and liberating nutrients and terminal electron acceptors[31–37]. These changes in the intestinal environment can be exploited by pathogens[22,32–36,38]. For example, utilisation of liberated amino acids and nitrate respiration provide *S*. Tm and *E. coli* with a competitive advantage over the microbiota[22,32–36]. Interestingly, systemic immune activation has been shown to cause transcriptomic and metabolic changes in the microbiota, suggesting that not only local but also extraintestinal inflammatory cues can affect the intestinal microenvironment[39]. Whether and how pathogens exploit gut-luminal changes upon systemic immune activation remains unknown to date.

Activation of the innate intestinal immune system triggers a spectrum of antimicrobial immune responses, including the release of antimicrobial peptides (AMP), molecular oxygen ($O_2$) or reactive oxygen species (ROS)[1,40–44]. Although these responses affect all microbiota species, some species have evolved mechanisms to cope with such host defence mechanisms[45,46]. The microbiota predominantly comprises obligate anaerobes, which are particularly sensitive to oxygen species ($O_2$ and ROS) as many of their enzymes contain iron-sulphur clusters that decompose upon oxidation, and damage cannot be repaired efficiently[47–49]. Facultative anaerobic pathogens such as *S*. Tm are less affected by oxygen species and can even exploit oxygen to their advantage, using it as a terminal electron acceptor for the electron transport chain, fuelling aerobic respiration[42,50,51]. In addition, the availability of terminal electron acceptors enables pathogens to utilise the oxidative tricarboxylic acid (TCA) cycle for efficient energy conservation, enabling them to rapidly outcompete the microbiota in oxygenated environments[52]. It remains unknown whether and how systemic immune activation impacts terminal acceptor availability in the gut lumen.

Systemic immune activation may perturb gut microbiota homeostasis, potentially driving opportunistic pathogen blooms, but the underlying mechanisms remain unclear. Here, we use a mouse model of acute, sublethal systemic LPS injection to achieve pathophysiological LPS concentrations and assess its impact on microbiota homeostasis. We show that systemic immune activation fosters gut-luminal blooms of *S*. Tm and common opportunistic pathogens through a transient increase in gut-luminal oxygen species, which inhibits microbiota fermentation and fuels gut-luminal pathogen blooms via oxidative respiration. These findings establish a direct link between acute immune activation and microbiota perturbations, revealing a mechanism by which transient inflammation disrupts gut homeostasis and favours pathogen overgrowth. This insight may inform interventions to mitigate infection risks in critically ill patients.

## Results

### Sublethal systemic LPS exposure enables gut-luminal pathogens to bloom in a TLR4-dependent manner

To study the effect of endotoxemia on gut microbiota homeostasis, we intravenously (i.v.) injected mice with a non-lethal dose of LPS (5 µg, ≈120 fold below the $LD_{50}$) (Fig. 1a). Injection of such low doses of LPS achieves serum LPS levels similar to those of human patients (2.5 µg LPS i.p. in mice yields ≈2 ng ml$^{-1}$ after 1 h and ≈0.4 ng ml$^{-1}$ 24 h)[53]. Importantly, this LPS dose activates systemic immune responses, while preserving epithelial integrity and barrier function[53]. To assess the impact of systemic immune activation on colonisation resistance, we subsequently orally infected mice with *S*. Tm (5 × 10$^7$ colony-forming unit (c.f.u.)) (Fig. 1a) or opportunistic pathogens. *S*. Tm is a well-characterised enteric pathogen that has been instrumental for the understanding of colonisation resistance using well-established mouse models[54,55]. In addition, the detailed knowledge about the pathogen, its physiology, and its genetic accessibility enable the molecular identification and dissection of pathways contributing to infection[56,57]. We use colonisation resistance against *S*. Tm as a sensitive readout for microbiota homeostasis. As expected, C57BL/6 mice harbouring an unperturbed complex microbiota were protected against gut-luminal blooms of *S*. Tm (Fig. 1b)[54]. In sharp contrast, systemic LPS exposure increased the faecal *S*. Tm density by 10'000-fold at 24 h post-infection (h.p.i.) compared to PBS-injected control mice (Fig. 1b). Strikingly, these faecal *S*. Tm loads were almost as high as in mice where we alleviated colonisation resistance by antibiotic pretreatment (i.e. streptomycin, Fig. 1b)[54]. Elevated faecal *S*. Tm loads were observed as of 12 h.p.i. and were independent of the tested inoculum size (Supplementary Fig. 1a, b). Notably, with a replication rate of approximately 7 generations per 24 h, the frequency with which *S*. Tm replicated in the gut lumen of LPS-exposed mice was similar to the one in antibiotic-pretreated mice, while *S*. Tm replication was much slower in PBS-injected mice (Fig. 1c). This shows that endotoxin-induced systemic immune activation breaks colonisation resistance against *S*. Tm.

In a variety of oral *S*. Tm infection models, the *S*. Tm type three secretion system 1 (TTSS-1) and its effectors (SopB, SipA, SopE, SopE2) mediate active tissue invasion, which elicits full-blown inflammation and fuels colonisation[32,58]. To assess the role of TTSS-1 effectors in the LPS model, we infected mice with a *S*. Tm mutant lacking SopB, SipA, SopE, and SopE2 (Δ4). Strikingly, LPS also boosted the gut-luminal bloom of this TTSS-1 virulence mutant by 1'000-fold (Fig. 1d), suggesting that blooming occurs independent of virulence. Furthermore, the gut-luminal *S*. Tm population in LPS-exposed mice showed limited expression of the TTSS-1 chaperone SicA, involved in the regulation of TTSS-1 effector proteins (Fig. 1e)[59]. Together, these data indicate that *S*. Tm does not rely on TTSS-1 virulence factors to bloom in the gut lumen post systemic immune activation. As opportunistic pathogens are frequently part of the gut microbiota and lack TTSS virulence factors[16,17], we then investigated whether systemic LPS exposure also opens a niche for clinically relevant opportunistic pathogens. Strikingly, systemic LPS exposure also enabled blooms of *E. coli* and

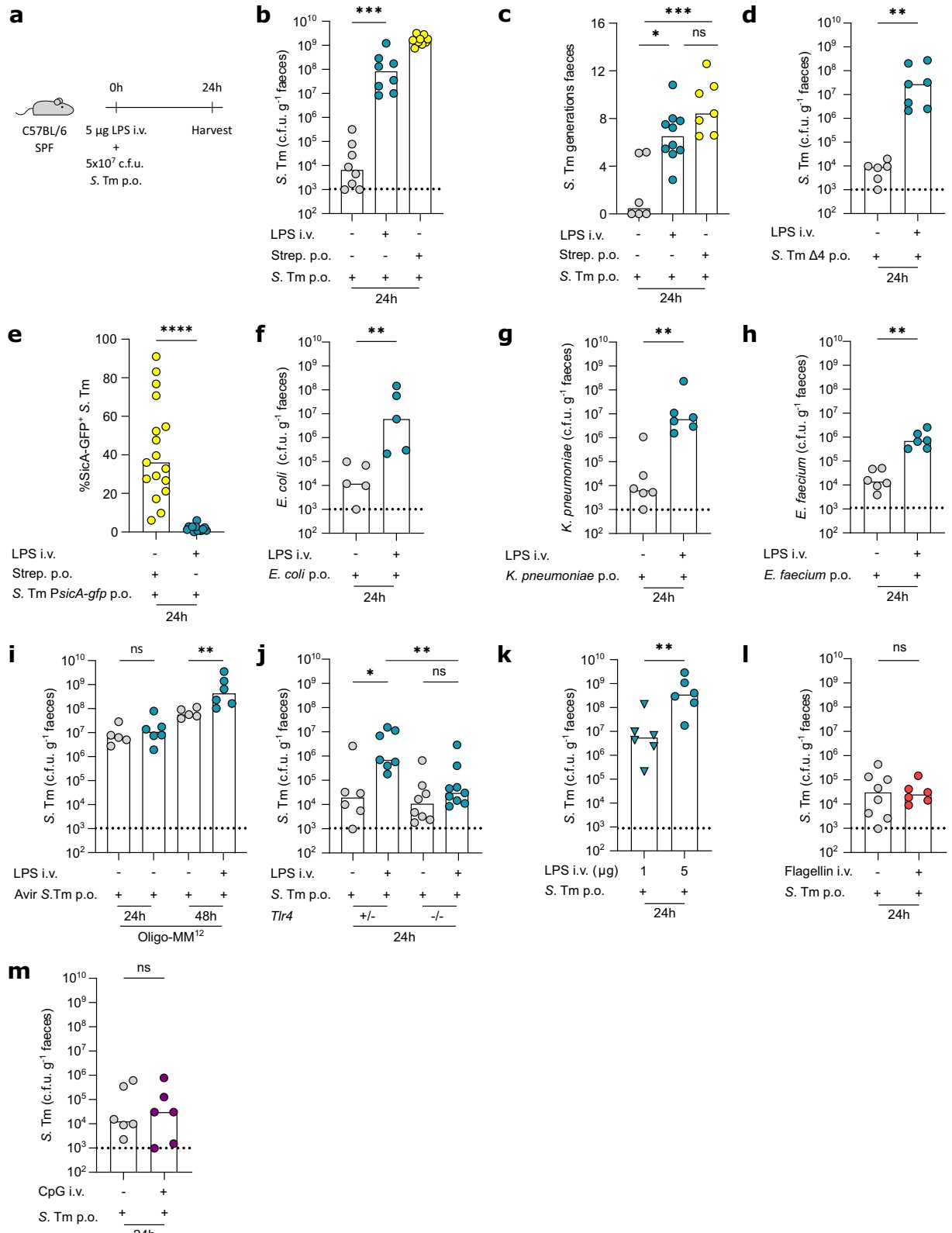

*K. pneumoniae*, two *Enterobacteriaceae* related to *S.* Tm, as faecal densities increased by 1'000-fold by 24 h.p.i. compared to PBS-injected mice (Fig. 1f, g). In addition, LPS exposure promoted a 100-fold increase in faecal loads of gram-positive opportunistic pathogen *E. faecium* (Fig. 1h). This shows that systemic immune activation drives the gut-luminal bloom of *S.* Tm and opportunistic pathogens independent of TTSS-1 virulence effector-mediated intestinal tissue

invasion and thereby uncouples gut-luminal pathogen blooms from classical pathogenic traits.

We next assessed whether increased gut-luminal loads facilitated systemic dissemination. Strikingly, systemic LPS exposure significantly increased *S.* Tm loads in the mesenteric lymph node (mLN) and spleen by 10- and 650-fold, respectively (Supplementary Fig. 1c, d). However, despite the bloom of *E. coli* in the gut lumen, LPS exposure did not

**Fig. 1 | Sublethal systemic LPS exposure promotes gut-luminal pathogens to bloom in a TLR4-dependent manner. a** C57BL/6 specific pathogen-free (SPF) mice were intravenously (i.v.) injected with 5 µg lipopolysaccharide (LPS) and orally infected with $5 \times 10^7$ c.f.u. *Salmonella* Typhimurium (*S.* Tm), faeces were collected 24 h post-infection (h.p.i.), unless indicated otherwise (**b**) Faecal *S.* Tm loads in mice systemically exposed to PBS or LPS or orally pre-treated with streptomycin (strep.) (minimum mice $n = 8$, at least two independent replicates, ***$P = 0.0002$). **c** Number of *S.* Tm replications in mice systemically exposed to PBS or LPS or orally pre-treated with strep. (minimum mice $n = 6$, at least two independent replicates, *$P = 0.0478$, ***$P = 0.0008$). **d** Faecal Δ*sopB* Δ*sipA* Δ*sopE* Δ*sopE2* (Δ4) *S.* Tm loads in mice systemically exposed to PBS or LPS (minimum mice $n = 6$, at least two independent replicates, **$P < 0.0012$). **e** Percentage caecal *S.* Tm expressing SicA in mice orally pre-treated with strep. or systemically exposed to LPS (minimum mice $n = 13$, at least two independent replicates, ****$P < 0.0001$). **f**–**h** Faecal *Escherichia coli*, *Klebsiella pneumoniae*, and *Enterococcus faecium* loads in mice systemically exposed to PBS or LPS (minimum mice $n = 5$, at least two independent replicates, **$P = 0.0079$, **$P = 0.0022$ **$P = 0.0022$). (**i**) Faecal *S.* Tm loads at 24 h and 48 h.p.i. in Oligo-MM[12] mice orally infected with avirulent *S.* Tm at 0 h.p.i. and systemically exposed to PBS or LPS at 24 h.p.i. (minimum mice $n = 5$, at least two independent replicates, **$P = 0.0087$). **j** Faecal *S.* Tm loads in *Tlr4*[+/-] and *Tlr4*[-/-] littermates systemically exposed to PBS or LPS (minimum mice $n = 6$, at least two independent replicates). **k** Faecal *S.* Tm loads in mice systemically exposed to 1 or 5 µg LPS (minimum mice $n = 6$, at least two independent replicates, **$P = 0.0043$). (**l**, **m**) Faecal *S.* Tm loads in mice systemically exposed to PBS or 12.5 µg flagellin or 1.8 µg ODN 2395 (minimum mice $n = 6$, two independent replicates). Bars indicate median values. Dotted lines indicate conservative average limit of detection. *P* values were calculated using the two-sided Mann-Whitney U test (**b**, **d**–**o**) or two-sided Kruskal-Wallis test with Dunn's multiple test correction (**c**). ns, not significant. Source data are provided in the Source Data file.

result in a corresponding increase in *E. coli* burden in the mLN or spleen (Supplementary Fig. 1e, f). This difference may be explained by the expression of TTSS virulence factors that support microbial survival of host systemic immune defences, which are present in *S.* Tm but absent in the *E. coli* strain used in this study. These findings suggest that while gut-luminal blooms of both *S.* Tm and opportunistic pathogens can occur independent of classical virulence factors, systemic dissemination may require additional virulence mechanisms.

To test whether the LPS-induced *S.* Tm bloom in the gut lumen depended on microbiota composition, we next infected Oligo-MM[12] mice, which harbour a defined microbiota that provides partial colonisation resistance against *S.* Tm[55]. LPS exposure increased the faecal *S.* Tm density 6-fold at 24 h.p.i. compared to PBS-injected Oligo-MM[12] mice (Supplementary Fig. 1g), indicating that the observed effect is relevant in different microbiota backgrounds. As the gut microbiota is a reservoir for opportunistic pathogens, we assessed the effect of systemic immune activation on an established gut-luminal avirulent *S.* Tm population, lacking the TTSS-1 and TTSS-2. This mutant behaves like a commensal *E. coli*, as it cannot actively spread beyond the gut lumen. The effect of systemic immune activation on a gut-luminal resident population of this mutant cannot be addressed in mice harbouring a complex microbiota, as they exhibit strong colonisation resistance against *S.* Tm (Fig. 1b). The Oligo-MM[12] model, by contrast, allows pre-colonisation with avirulent *S.* Tm (Fig. 1i), enabling us to study the effect of systemic immune activation on a gut-luminal reservoir of avirulent *S.* Tm. I.v. injection of LPS 24 h.p.i., when the faecal avirulent *S.* Tm load was ~$10^7$ c.f.u. g⁻¹ faeces, boosted faecal avirulent *S.* Tm densities 10-fold by 48 h.p.i., compared to PBS-injected mice (Fig. 1i). Pre-colonisation with wild-type *S.* Tm instead of avirulent *S.* Tm and subsequent LPS-exposure led to a similar 10-fold increase by 48 h.p.i (Supplementary Fig. 1h). This confirms that acute systemic immune activation does not only enable initial overgrowth but also niche expansion of gut lumen-resident virulent and avirulent *S.* Tm populations.

LPS is a pathogen-associated molecular pattern (PAMP) that triggers an inflammatory cascade upon sensing by host cells via Toll-like receptor 4 (TLR4) or Caspase-4/11[60–62]. Infection of *Tlr4*[-/-] and *Casp11*[-/-] mice showed that extracellular LPS sensing via TLR4 is required in our model (Fig. 1j, Supplementary Fig. 1i). In addition, the *S.* Tm bloom scaled with the LPS dose (Fig. 1k). This suggests that TLR4-mediated systemic immune activation causes a tunable response that opens a niche for gut-luminal *S.* Tm blooms. Flagellin and bacterial DNA containing unmethylated CpG motifs are alternative PAMPs that also trigger inflammatory responses, albeit via different receptors[41,63]. We, therefore, assessed whether systemic innate immune activation with these PAMPs also enables *S.* Tm to bloom. Notably, the effect was PAMP-specific, but not species-specific, as exposure of mice to equimolar doses of flagellin or CpG did not enable *S.* Tm to bloom in the gut lumen (Fig. 1l, m), while exposure to both *S.* Tm and *E. coli* LPS

increased faecal *S.* Tm loads (Fig. 1b, Supplementary Fig. 1j). Thus, LPS sensing via TLR4, but not flagellin or CpG, triggers a transient response that enables *S.* Tm to bloom in the intestine. Taken together, TLR4-mediated systemic immune activation establishes and expands a gut-luminal niche for *S.* Tm and opportunistic pathogens.

## Sublethal systemic LPS exposure triggers a transient intestinal inflammatory response that elevates gut-luminal oxygen species levels

Intestinal inflammation promotes gut-luminal blooms[32,33]. To assess whether systemic immune activation triggers intestinal inflammation that drives pathogen blooms, we measured the gut-luminal concentration of the inflammation marker lipocalin-2 and analysed mucosal histopathology. Lipocalin-2 levels increased in *S.* Tm-infected LPS-exposed mice compared to PBS-injected mice at 24 h.p.i., but were 500-fold lower than in *S.* Tm-infected antibiotic-pretreated mice (Fig. 2a). Furthermore, lipocalin-2 levels remained below $10^3$ ng g⁻¹ caecal content in LPS-exposed mice (Fig. 2a), which is considered a threshold for overt tissue-disruptive intestinal inflammation in the context of acute *S.* Tm infection in mice[64]. We also did not observe histological signs of caecal pathology indicative of acute *S.* Tm enterocolitis during the initial 24 h.p.i. (Fig. 2b)[32,54]. Overall, these data suggest that the LPS-triggered *S.* Tm bloom occurs independent of overt intestinal inflammation (Fig. 2a, b). This was particularly remarkable since *S.* Tm reached similar colonisation levels in LPS-exposed mice as in the classical streptomycin pretreatment model for gut infection (Fig. 1b), where gut colonisation by WT *S.* Tm typically goes along with overt gut inflammation (Fig. 2a, b).

We hypothesised that endotoxin exposure triggers a short intestinal inflammatory primer, rather than sustained, overt inflammation, that transiently perturbs the microbiota and thereby opens a niche in which gut-luminal pathogens can bloom. To evaluate this, we assessed the inflammatory response in non-infected LPS-exposed mice during the initial 12 h post-injection (h.p.inj.). Faecal lipocalin-2 significantly increased by 6 to 9 h post-LPS exposure (Fig. 2c), confirming that systemic LPS exposure induces a transient intestinal inflammatory primer that does not develop into tissue-disruptive overt inflammation (i.e. lipocalin-2 < $10^3$ ng g⁻¹). To gain a more detailed understanding of this inflammatory pulse, we performed bulk RNA sequencing (RNAseq) and flow cytometry analysis of the caecum tissue of LPS-exposed mice. Gene set over-representation analysis showed upregulation of genes involved in responses to molecules of bacterial origin, cytokine responses, and regulation of immune responses at 6 h post-LPS exposure (Fig. 2d). This included the upregulation of classical inflammatory response genes, such as *Tnf*, *Il22*, *Nos2*, *Cxcl9* and *Cxcl10*, which was confirmed by qPCR (Fig. 2e, Supplementary Fig. 2a). Assessment of immune cell populations by flow cytometry revealed that the number of macrophages, monocytes and eosinophils remained unchanged, whereas the number of neutrophils increased 10-fold in the caecum of

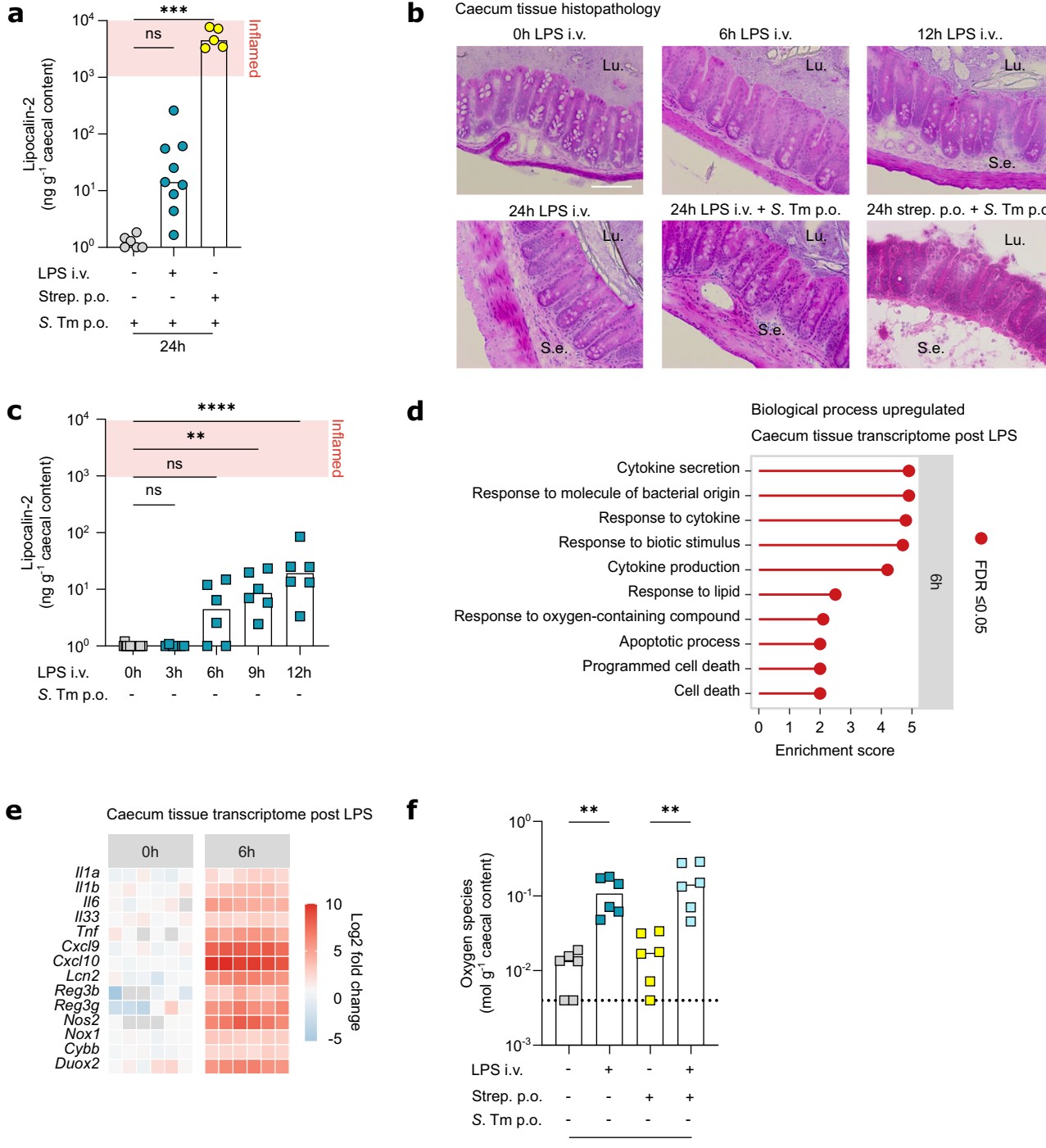

**Fig. 2 | Systemic LPS exposure triggers a transient inflammatory response that elevates gut-luminal oxygen species levels. a** Caecal lipocalin-2 levels at 24 h.p.i. in mice systemically exposed to PBS or LPS or orally pre-treated with streptomycin (minimum mice $n = 5$, at least two independent replicates, ***$P = 0.0002$). **b** Representative images of haematoxylin and eosin-stained caecum tissue at 0, 6, 12 or 24 h post-injection (h.p.inj) or 24 h.p.i. in mice systemically exposed LPS or orally pre-treated with streptomycin. Scale bar=80 μm (mice $n = 6$, at least two independent replicates). **c** Caecal lipocalin-2 levels at 0, 3, 6, 9 or 12 h.p.inj. in mice systemically exposed to LPS (minimum mice $n = 6$, at least two independent replicates, **$P = 0.0013$, ****$P < 0.0001$). **d** Caecum tissue RNAseq at 6 h.p.inj. in mice systemically exposed to LPS compared to PBS. Over-representation analysis of

biological processes based on significantly upregulated genes (minimum mice $n = 6$). **e** Caecum tissue RNAseq log2 fold change of classical bacterial response genes at 0 and 6 h.p.inj. in mice systemically exposed to LPS compared to PBS (minimum mice $n = 6$). **f** Caecal oxygen species levels at 3 h.p.inj. in mice systemically exposed to PBS or LPS, treated with or without streptomycin (minimum mice $n = 6$, at least two independent replicates, **$P = 0.0022$)· Bars indicate median values. Dotted lines indicate conservative average limit of detection. $P$ values were calculated using the two-sided Kruskal-Wallis test with Dunn's multiple test correction (**a,c**) or two-sided Mann-Whitney U test (**f**). ns, not significant. Source data are provided in the Source Data file.

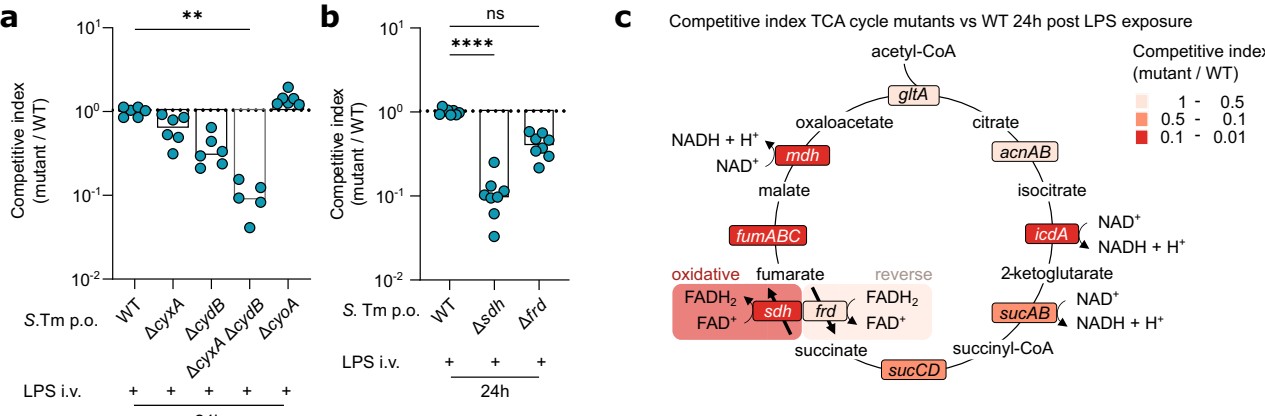

**Fig. 3 | S. Tm employs aerobic respiration and the oxidative TCA cycle to bloom in the gut of LPS-exposed mice. a** Competitive index of Δ*cyxA*, Δ*cydB*, Δ*cyxA* Δ*cydB*, and Δ*cyoA S.* Tm in 1:1 competition with WT *S.* Tm at 24 h.p.i. in mice systemically exposed to LPS (minimum mice *n* = 5, at least two independent replicates, **P = 0.0087). **b** Competitive index of Δ*sdh* and Δ*frd S.* Tm in 1:1 competition with WT *S.* Tm at 24 h.p.i. in mice systemically exposed to LPS (minimum mice *n* = 8, at least two independent replicates, ****P < 0.0001). **c** Competitive index of Δ*gltA*, Δ*acnAB*, Δ*icdA*, Δ*sucAB*, Δ*sucCD*, Δ*frd*, Δ*sdh*, Δ*fumABC* and Δ*mdh S.* Tm in 1:1 competition with WT *S.* Tm at 24 h.p.i. in mice systemically exposed to LPS (minimum mice *n* = 4, at least two independent replicates). Bars indicate median values. Dotted lines indicate competitive index of 1. *P* values were calculated using the two-sided Kruskal-Wallis test with Dunn's multiple test correction (**a**,**b**). ns, not significant. Source data are provided in the Source Data file.

LPS-exposed mice compared to PBS-injected mice (Supplementary Fig. 2b). However, using antibody depletion and knockout mice, we refuted a dependence of the LPS-induced *S.* Tm bloom on neutrophils, macrophages, B cells, T cells, NK cells, IFNAR, TNF, IL-22, and iNOS (Supplementary Fig. 2c–i). Altogether, we show that multiple classical inflammatory/antibacterial mediators are not required to mediate the LPS-induced *S.* Tm bloom, at least when ablated individually.

Interestingly, we detected upregulation of genes involved in responses to oxygen-containing compounds at 6 h.p.inj., including upregulation of intestinal NADPH oxidases *Nox1*, *Cybb*, and *Duox2* (Fig. 2d, e). In line with this, gut-luminal oxygen species levels, including O₂ and ROS, were increased in the caecum of LPS-exposed mice in a TLR4-dependent manner (Fig. 2f, Supplementary Fig. 2j). The oxygen species were likely host-derived, and not microbiota-derived, as we observed a similar increase in oxygen species upon LPS exposure in non-infected mice treated with antibiotics (Fig. 2f). Infection of *Cybb⁻/⁻* and *Nox1⁻/⁻* mice showed that the LPS-induced *S.* Tm bloom does not solely depend on CYBB or NOX1 (Supplementary Fig. 2k, l), suggesting potential redundancy in ROS generation pathways. In addition, LPS might cause an inflammation-elicited switch to lactate fermentation in the intestinal tissue, which reduces epithelial oxygen consumption and thereby increases diffusion of O₂ from the circulation into the gut lumen[65]. Importantly, flagellin injection neither induced *S.* Tm blooms (Fig. 1l), nor increased gut-luminal oxygen species levels (Supplementary Fig. 2m), despite the induction of a pro-inflammatory intestinal response comparable to that in LPS-exposed mice (Supplementary Fig. 2a). This refutes a general inflammation-induced phenotype, and rather points towards a causal role of LPS-elicited oxygen species in mediating blooms of *S.* Tm and opportunistic pathogens in our model. In conclusion, sensing of LPS via TLR4 induces a transient intestinal inflammatory response which triggers the release of oxygen species into the gut lumen.

### S. Tm employs aerobic respiration and the oxidative TCA cycle to bloom in the gut lumen of LPS-exposed mice

We next investigated whether LPS-induced oxygen species enable the observed *S.* Tm bloom in our model by providing a competitive advantage for *S.* Tm over the microbiota. *S.* Tm encodes three cytochrome oxidases, which are employed for aerobic respiration at distinct levels of oxygen saturation[50]. *S.* Tm utilises cytochrome bd-II oxidase (CyxAB) for growth under low oxygen conditions (0.8% oxygen), cytochrome bd oxidase (CydAB) under intermediate oxygen levels (3–8% oxygen), and cytochrome bo3 (CyoAB) under full aeration[50]. To assess whether *S.* Tm exploits gut-luminal oxygenation in our model, we infected LPS-injected mice with a mix of wildtype (WT), Δ*cyxA*, Δ*cydB*, Δ*cyoA*, and Δ*cyxA* Δ*cydB S.* Tm. Δ*cydB S.* Tm had a fitness defect compared to WT *S.* Tm (Fig. 3a). There was also a trend for lower fitness of the Δ*cyxA* mutant, while Δ*cyoA* was not affected (Fig. 3a). Strikingly, a *S.* Tm double mutant lacking both *cyxA* and *cydB* was 10-fold attenuated compared to WT *S.* Tm in LPS-exposed mice (Fig. 3a), while *S.* Tm does not depend on cytochrome oxidases in untreated, naïve mice harbouring an unperturbed complex gut microbiota[50]. This indicates that aerobic respiration contributes to gut-luminal *S.* Tm expansion post-LPS exposure.

The TCA cycle plays an essential role in the generation of energy and precursors for biosynthesis pathways, via the breakdown of carbon compounds and the subsequent transfer of high-energy electrons to the electron transport chain. Availability of oxygen as a terminal electron acceptor for the electron transport chain enables *S.* Tm to complete the oxidative TCA cycle[52]. Succinate dehydrogenase (Sdh) catalyses the oxidation of succinate to fumarate in the TCA cycle. In the absence of oxygen, *S.* Tm does not express Sdh; instead, it reduces fumarate to succinate via fumarate reductase (Frd), meaning that this part of the TCA cycle runs in reverse under anaerobic conditions[66–68]. The dependence on Sdh instead of Frd can therefore serve as an indicator of aerobic growth[52,66–68]. We infected mice with a mix of WT, Δ*sdh* and Δ*frd S.* Tm, to determine whether *S.* Tm utilises the oxidative or reverse TCA cycle to bloom in the gut lumen of mice exposed to LPS. The Δ*sdh S.* Tm mutant was 10-fold attenuated compared to WT *S.* Tm, while the Δ*frd S.* Tm mutant was attenuated to a lesser extent (Fig. 3b). This, together with dependence on the terminal oxidases CyxA and CydB, suggests that *S.* Tm employs the oxidative TCA cycle, and therefore aerobic growth, to fuel the observed bloom. *S.* Tm mutants lacking further TCA cycle genes, such as Δ*mdh* and Δ*fumABC*, had a competitive disadvantage of 5- to 20-fold (Fig. 3c), confirming the importance of the TCA cycle for *S.* Tm growth in our model. Altogether, we show that *S.* Tm relies on aerobic respiration together with the oxidative TCA cycle to fuel the gut-luminal bloom upon systemic LPS exposure. As *E. coli* and *K. pneumoniae* are facultative anaerobes that encode for similar aerobic respiration pathways as *S.* Tm[69], these opportunistic pathogens can potentially employ similar metabolic strategies to bloom in the gut lumen of LPS-exposed mice.

## Systemic immune activation exposes the microbiota to oxidative stress, inhibiting microbiota fermentation

As a large fraction of the microbiota consists of obligate anaerobes[49], we hypothesised that the LPS-triggered increase in gut-luminal oxygen species impairs microbiota homeostasis. Surprisingly, the plateable anaerobic caecal microbiota density was not affected by LPS injection (Fig. 4a). Furthermore, LPS exposure did not affect microbiota richness and evenness up to 24 h.p.inj. of mice harbouring a complex microbiota or a gnotobiotic Oligo-MM[12] microbiota, as determined by 16S rRNA gene sequencing (Fig. 4b, Supplementary Fig. 3a, b). This indicates that systemic LPS exposure does not alter microbiota density and composition at 24 h.p.inj., at least in mice lacking facultative anaerobe opportunistic pathogens like *E. coli* or *K. pneumoniae* in their microbiota (Fig. 4a, b, Supplementary Fig. 3a, b).

To gain an understanding of how LPS exposure affects the microbiota, we performed metatranscriptome analysis of the microbiota of LPS-exposed mice. We used Oligo-MM[12] mice, as the genomes of the 12-microbiota member community are known, enabling metatranscriptome analysis at strain resolution[55]. LPS exposure of Oligo-MM[12] mice induced upregulation of genes involved in responses to oxidative stress, including ROS detoxification (*sod2, ahpF*) and iron-sulphur cluster assembly (*iscU*), which help to replace damaged electron transport chain enzymes, as well as chaperones that guide protein folding (*clpB, dnaJ, grpE, hslO*), which are particularly relevant during oxidative stress (Fig. 4c, Supplementary Fig. 3c)[70]. This confirms that systemic LPS exposure triggers a transcriptional oxidative stress response in the caecal microbiota. Strikingly, microbiota members downregulated genes encoding for ribosomal proteins (*rpl3, rps10*) and involved in amino acid biosynthesis (*leuB, hisB*) (Fig. 4c, Supplementary Fig. 3c). As ribosomes and amino acids are essential for bacterial translation, this indicates a reduction in microbiota protein synthesis[71]. Taken together, these data support that the microbiota of LPS-exposed mice experiences an oxidative stress-induced shift from microbiota protein synthesis to protein repair. This provides evidence for reduced microbiota metabolism upon acute systemic LPS exposure.

The microbiota ferments dietary substrates, such as complex carbohydrates and amino acids[27–30]. To assess how transcriptional responses relate to gut-luminal metabolite levels as a readout of microbiota metabolism, we quantified gut-luminal metabolites by targeted metabolomics post-LPS exposure. Extracellular amino acid levels in the caecum lumen were higher in untreated germ-free mice, which lack a microbiota, than in mice harbouring a complex microbiota (Fig. 4d, Supplementary Fig. 4a), confirming that the microbiota consumes amino acids[22]. Strikingly, LPS exposure significantly elevated the levels of the amino acids L-aspartate and L-glutamate in mice harbouring a complex microbiota, but not in germ-free mice (Fig. 4d, Supplementary Fig. 4a, b). This suggests that utilisation of these amino acids by the microbiota is halted upon systemic immune activation[22]. We further assessed microbiota activity by measuring their fermentation products, SCFAs, and hydrogen ($H_2$)[72]. LPS exposure significantly decreased caecal SCFAs acetate, propionate, and butyrate levels (Fig. 4e–g). $H_2$ production by the complex microbiota is readily detected in ambient gas when using metabolic cages, and LPS exposure reduced $H_2$ production in mice from 2 to 12 h.p.inj. (Fig. 4h, Supplementary Fig. 4c). This indicates that gut microbiota fermentation is negatively affected in LPS-exposed mice. Importantly, LPS-exposed mice also reduced food intake, which limits the amount of fermentation substrates available for the microbiota (Supplementary Fig. 4d). However, in PBS-injected control mice, $H_2$ production did not strongly correlate with food consumption ($R^2 = 0.16$; Supplementary Fig. 4e), indicating that the reduction in $H_2$ production in LPS-exposed mice can occur independently of the reduction in food consumption. Taken together, this confirms that systemic immune activation triggers a microbiota stress response that inhibits fermentation by the resident microbiota.

## Oxidative stress inhibits microbiota fermentation and growth, promoting facultative anaerobic pathogens to bloom

In vitro studies have shown that some obligate anaerobic bacteria are able to survive transient oxygen exposure and resume growth after the damage has been repaired[73]. We hypothesised that pathogens may bloom after gut-luminal oxygen species exposure if their recovery is faster and when they can utilise oxygen for energy conservation (Fig. 3a–c), as previous studies showed that *S.* Tm can utilise oxygen to bloom during intestinal infections[50,52]. In these studies, antibiotic treatment reduced butyrate-producing microbiota members and shifted intestinal epithelial metabolism from butyrate oxidation to lactate fermentation, causing excess oxygen to leak into the gut lumen and enabling oxidative respiration of pathogens[50,51,65]. In this context, oxygen species function as a consequence of microbiota perturbations rather than a primary cause. To assess a potential causal link between oxidative stress and an arrest in microbiota growth in the absence of further host effects and in a controllable system, we turned to in vitro assays that simulate transient spikes in oxygen species levels. Specifically, we used $H_2O_2$, a ROS commonly produced in the context of inflammation and antibacterial responses[74]. While $H_2O_2$ concentrations measured in the gut lumen under inflammatory conditions are typically in the μM range (i.e. a fraction of the ≈100 μM oxygen species measured in the cecum lumen of LPS-exposed mice) (Fig. 2), we sought to investigate how transient, localised bursts of $H_2O_2$, potentially reaching mM range close to the tissue, could impact microbial growth. To this end, we exposed common human gut microbiota strains[75], Oligo-MM[12] strains[55], *S.* Tm, and the opportunistic pathogens tested in LPS-exposed mice to different $H_2O_2$ concentrations. We determined the minimum $H_2O_2$ concentrations inhibiting the growth of the respective strain by at least 25 or 75% (IC25, IC75). It should be noted that while the indicated concentrations refer to the target concentrations of $H_2O_2$ added to the respective wells, the actual $H_2O_2$ concentrations are likely lower due to reactions of $H_2O_2$ with the anaerobic compounds present in the buffer (see data below; Supplementary Fig. 5a, b). Strikingly, we observed that most tested strains had a similar IC25 of 1.5–3 mM $H_2O_2$ (Fig. 5a). However, *S.* Tm, *E. coli, K. pneumoniae*, and *E. faecium* were significantly more resistant than most of the commensal strains tested, with an IC75 of 6 mM compared to 1.5-3 mM $H_2O_2$ (Fig. 5a–c). Thus, facultative anaerobes that bloomed in the gut lumen of LPS-exposed mice, have a higher tolerance to oxidative stress than common commensal microbiota strains.

Next, we assessed how a complex human microbiota, which naturally contains opportunistic pathogens, responds to oxidative stress, using the well-established Polyfermentor Intestinal Model (PolyFermS)[76–78]. In this set-up, a first-stage inoculation reactor was seeded with immobilised faecal microbiota, and then used to inoculate parallel test reactors (TR) to establish a stable and reproducible human microbiota undergoing continuous anaerobic fermentation (Supplementary Fig. 5c). To determine whether *S.* Tm outcompetes commensal microbiota members post oxidative stress, we inoculated the TRs with *S.* Tm. 12 h post-*S.* Tm inoculation, the TR content was exposed to a final concentration of 0 (TR1), 1.5 (TR2), 3 (TR3) or 8 mM (TR4) $H_2O_2$, based on the IC75 observed in the single strain in vitro experiments (Fig. 5a–c, Supplementary Fig. 5a–c). As noted earlier, the effective $H_2O_2$ concentrations decline rapidly after its addition, presumably due to reactions with media components and bacterial compounds (Supplementary Fig. 5a, b,). We aimed to recapitulate the colonisation and transient LPS-elicited oxygen species release in the pre-colonisation experiments in Oligo-MM[12] mice (Figs. 1i, 2f, Supplementary Fig. 1h). Metabolite levels, *S.* Tm densities and microbiota composition were analysed in the TR effluent for 72 h. Notably, the in vitro human microbiota model faithfully recapitulated our in vivo

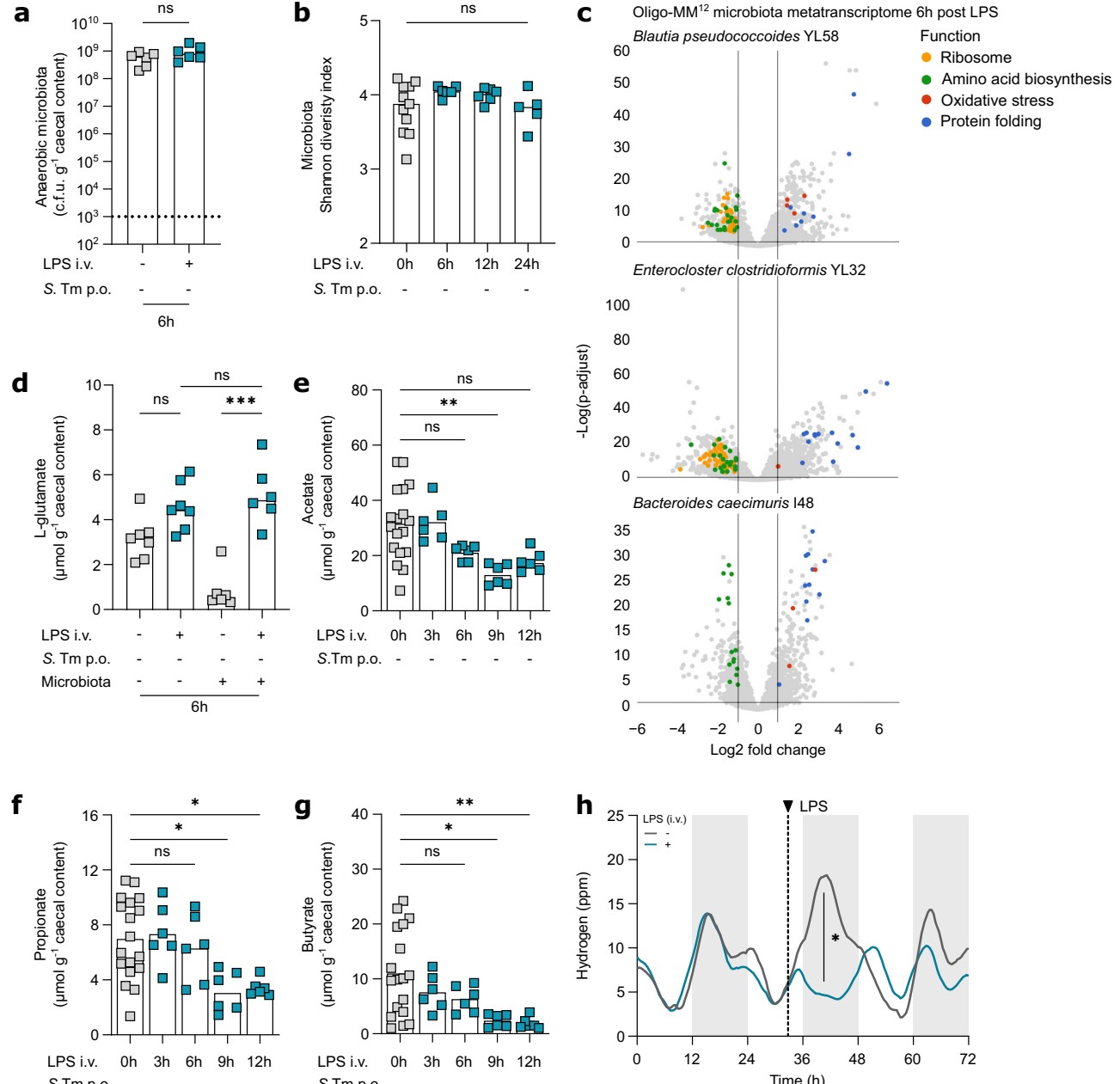

**Fig. 4 | Systemic LPS exposure triggers a microbiota stress response that inhibits microbiota fermentation. a** Caecal content anaerobic microbiota loads at 6 h.p.inj. in mice systemically exposed to PBS or LPS (mice $n = 6$, at least two independent replicates). **b** 16S rRNA gene sequencing Shannon diversity index at 0, 6, 12 or 24 h.p.inj. in mice systemically exposed to LPS (minimum mice $n = 5$ at least two independent replicates). **c** Metatranscriptome analysis of the caecal microbiota at 6 h.p.inj. in Oligo-MM[12] mice systemically exposed to LPS compared to PBS, representative volcano plots of gene distribution of *Blautia pseudococcoides* YL58, *Enterocloster clostridioformis* YL32 and *Bacteroides caecimuris* I48 (minimum mice $n = 6$, at least two independent replicates). **d** Caecal L-glutamate levels in germ-free and SPF mice systemically exposed to PBS or LPS at 6 h.p.inj. (minimum mice $n = 6$, at least two independent replicates, ***$P = 0.0010$). **e–g** Caecal acetate, propionate,

and butyrate levels at 0, 3, 6, 9 or 12 h.p.inj. in mice systemically exposed to LPS (minimum mice $n = 6$, at least two independent replicates, acetate: **$P = 0.0034$, propionate: 0 h vs 9 h *$P = 0.0446$ and 0 h vs 12 h *$P = 0.0372$, butyrate: *$P = 0.0309$ and **$P = 0.0095$). **h** Hydrogen levels of mice systemically exposed to PBS or LPS at 32 h, curves obtained by smoothing function of data obtained every 24 min per mouse (mice $n = 4$, at least two independent replicates, *$P = 0.0286$). Bars indicate median values. Dotted lines indicate conservative average limit of detection. Dashed lines indicate time of injection. Grey rectangles indicate dark phase. $P$ values were calculated using the two-sided Mann-Whitney U test (**a,h**), two-sided Kruskal-Wallis test with Dunn's multiple test correction (**b,d-g**) or two-sided Wald test with Benjamini-Hochberg multiple test correction (**c**). ns, not significant. Source data are provided in the Source Data file.

observations, including a transient increase in amino acid concentrations (L-aspartate, L-glutamate), and a transient decrease in SCFAs acetate, propionate, and butyrate in the TRs post-$H_2O_2$ exposure (Fig. 6a–d, Supplementary Fig. 6a–d). This indicates that a spike of oxidative stress is sufficient to transiently reduce fermentation by a complex human microbiota and that the microbiota can recover from this transient challenge to regain normal function.

In line with the observations in the pre-colonisation model in Oligo-MM[12] mice (Fig. 1i, Supplementary Fig. 1h), *S*. Tm bloomed upon $H_2O_2$ exposure (Fig. 6e, Supplementary Fig. 6e). In addition, the $H_2O_2$ pulse enabled facultative anaerobic opportunistic pathogen microbiota members from the *Enterococcaceae* family (TR3, 3 mM) and from the *Enterobacteriaceae* family (TR4, 8 mM) to bloom, peaking at 12 h post-$H_2O_2$ exposure (up to $10^9$ c.f.u. ml⁻¹; black boxes in Fig. 6f,

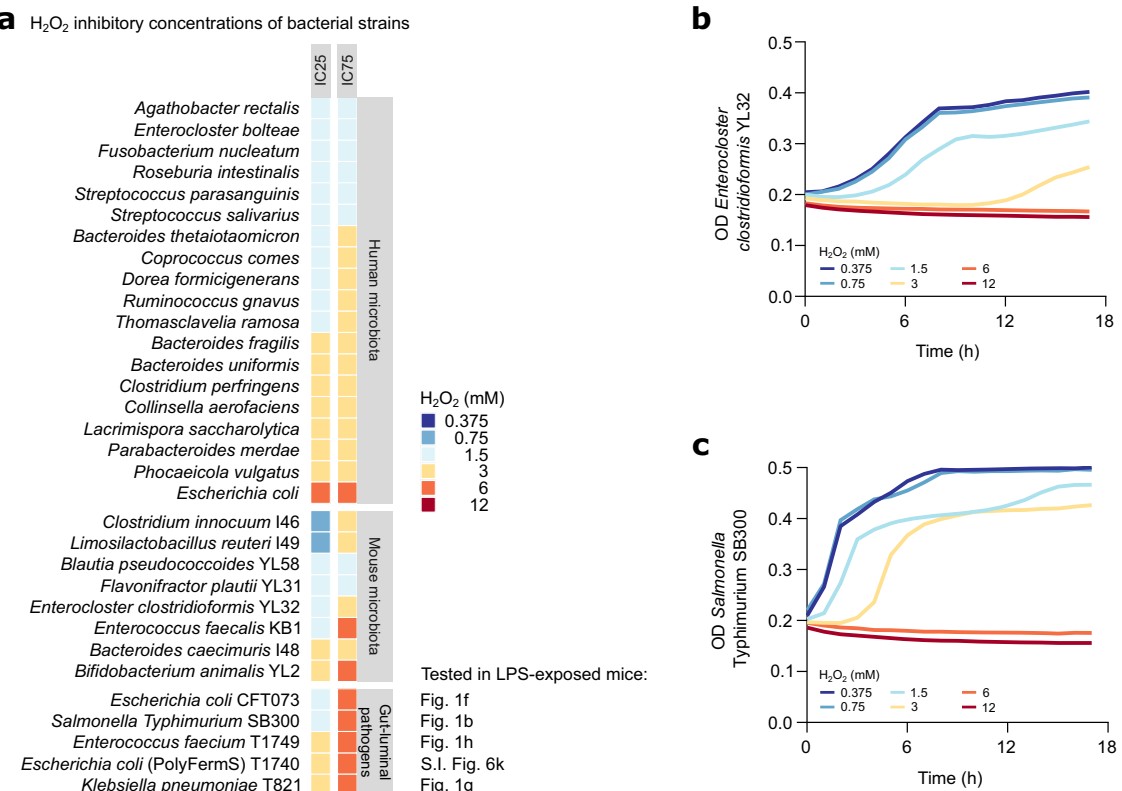

**Fig. 5 | Facultative anaerobic pathogens exhibit a higher tolerance to oxidative stress than common commensal microbiota strains. a–c** Bacterial strains grown twice overnight and adjusted to a starting OD of 0.05 were exposed to $H_2O_2$ concentrations ranging from 0.375 to 12 mM. Growth curves were measured and inhibitory concentrations 25 and 75 (IC25, IC75) were calculated (wells $n = 4$). Source data are provided in the Source Data file.

Supplementary Fig. 6f–j). Strikingly, this growth was even more pronounced than that of *S.* Tm, which merely reached up to $10^5$ c.f.u. ml$^{-1}$ (Fig. 6e). This shows that also in a human microbiota, facultative anaerobes, such as *S.* Tm, *E. faecium* and *E. coli*, can outcompete obligate anaerobes post-oxidative stress and suggests that different facultative anaerobes compete for the same opened niche. It further shows that stochastic factors can allow the outgrowth of different competitors, as exemplified in TR3 (3 mM $H_2O_2$) in which *Enterococcaceae* bloomed rather than *S.* Tm (Fig. 6e, f). Importantly, in contrast to the human gut microbiota consortium employed in the in vitro model, the microbiota of the mice used in Figs. 1–4 is devoid of facultative anaerobic opportunistic pathogens. In consortia lacking such competitors, *S.* Tm can reach densities of $10^7$–$10^9$ c.f.u. g$^{-1}$ faeces (Fig. 1b). To assess whether the facultative anaerobic opportunistic pathogens that bloomed in the in vitro human gut microbiota model could also bloom in the intestine of LPS-exposed mice, we isolated the predominant *E. faecium* and *E. coli* strains post-$H_2O_2$ exposure from TR3 (3 mM $H_2O_2$) and TR4 (8 mM $H_2O_2$), respectively, and infected mice with these isolates. Strikingly, LPS exposure increased faecal *E. faecium* loads 100-fold and faecal *E. coli* (PolyFermS) loads 1'000-fold (Fig. 1f, Supplementary Fig. 6k). Collectively, these observations are in line with the observations in the LPS mouse model and support that a short pulse of oxidative stress alone is sufficient to promote facultative anaerobic gut-luminal pathogens to bloom in response to microbiota inhibition in a human context.

## Discussion

The acute phase of critical illness frequently coincides with an acute increase in plasma LPS levels and gut-luminal opportunistic pathogen blooms, however, a mechanistic link between these phenomena remained obscure. Here, we used a murine model of acute endotoxin exposure to assess how systemic immune activation affects the susceptibility to gut-luminal pathogen blooms. Using host and microbiota transcriptomics and metabolomics in combination with mouse and human microbiota models, we mechanistically resolve an important link between acute endotoxin-induced systemic immune activation, oxygen species-mediated gut microbiota inhibition, and gut-luminal pathogen blooms via oxidative respiration. Importantly, in contrast to previous reports focusing on the inflamed gut of infected mice[32,54], here we show that this can occur independently of overt intestinal inflammation (e.g. triggered by bacterial virulence factors) and drastic shifts in microbiota composition (e.g., upon antibiotic treatment) and that this mechanism is relevant for human microbiota consortia.

Systemic immune activation has previously been demonstrated to trigger transcriptional and metabolic shifts in the microbiota[39], but the underlying mechanisms and implications for the host remained unclear. We here uncover that acute systemic immune activation primes a transient inflammatory pulse in the intestine, which leads to the release of oxygen species (i.e. ROS or $O_2$) into the gut lumen. Using extensive phenotyping of microbiota and host responses, we further show that this induces an oxidative stress response in oxygen-sensitive microbiota members, inducing a shift from protein synthesis to protein repair and reducing microbiota fermentation. This represents a mechanism by which systemic immune activation perturbs the intestinal microenvironment.

We further demonstrate that clinically relevant opportunistic pathogens and *S.* Tm can exploit changes in the intestinal microenvironment post systemic immune activation. Previous studies linked low microbiota complexity, either at baseline or induced via antibiotic treatment or overt intestinal inflammation, to gut-luminal pathogen blooms[32,33,50,54]. Strikingly, we here demonstrate that a short inflammatory pulse alone is sufficient to transiently inhibit microbiota

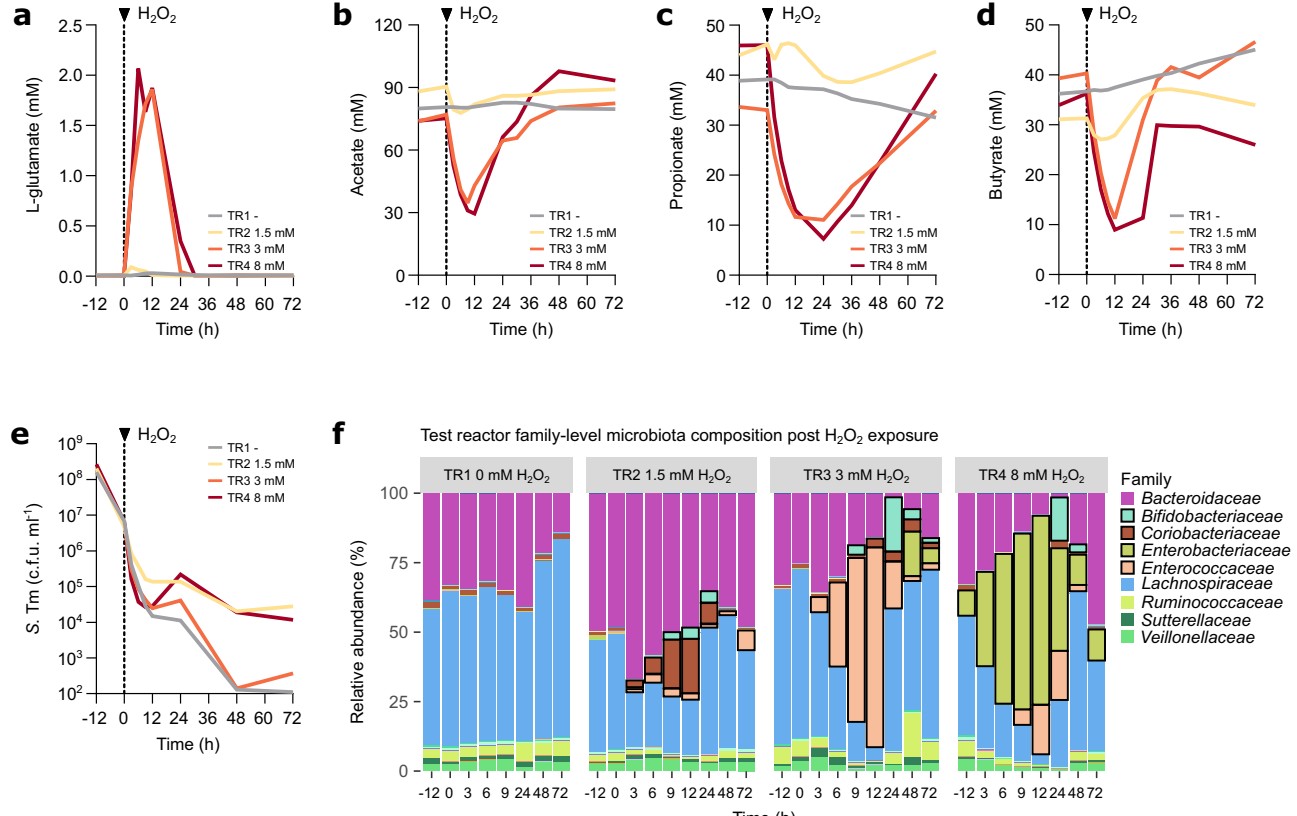

**Fig. 6 | Oxidative stress inhibits human gut microbiota fermentation and promotes facultative anaerobic pathogens to bloom. a–d** An inoculation reactor containing immobilised faecal microbiota of an adult donor inoculated test reactors (TR1-4). TRs were inoculated with *S.* Tm at −12h to reach a density of $1 \times 10^8$ c.f.u. ml⁻¹. 12 h post *S.* Tm inoculation, TRs were exposed to varying H₂O₂ concentrations 0 mM (TR1), 1.5 mM (TR2), 3 mM (TR3), 8 mM (TR4) (reactors $n = 4$). TR effluent was collected every 3 h to 12 h. L-glutamate, acetate, propionate, and butyrate concentrations in TR effluent post-H₂O₂ exposure (reactors $n = 4$). **e** *S.* Tm loads in TR effluent post-H₂O₂ exposure (reactors $n = 4$). **f** 16S rRNA gene sequencing family-level microbiota composition in TR effluent post-H₂O₂ exposure (reactors $n = 4$). Dashed lines indicate time of H₂O₂ exposure. Source data are provided in the Source Data file.

function, enabling gut-luminal pathogen blooms independently of lasting alterations in microbiota composition. Notably, gut-luminal pathogen colonisation reaches almost the same levels as in antibiotic-pretreated mice, a prerequisite for subsequent virulence factor-induced inflammation in several commonly used models for *S.* Tm infection[32,54]. In our model, the gut-luminal bloom is independent of invasion-related virulence factors and it relies on the difference between obligate and facultative anaerobes in the ability to cope with oxygen species (O₂, H₂O₂, and other ROS) and exploit aerobic respiration. We show that facultative anaerobes are less sensitive when exposed to oxygen species, and that the competitive advantage of facultative anaerobes over obligate anaerobes hinges on the ability to switch to aerobic respiration and the oxidative TCA cycle. Previous studies showed that antibiotic treatment reduces butyrate-producing microbiota members and shifts intestinal epithelial metabolism from butyrate oxidation to lactate fermentation. This causes excess oxygen to leak into the gut lumen and enables oxidative respiration of pathogens[50,51,65]. Importantly, we here demonstrate that oxygen species can not only be a consequence, but also a cause of gut microbiota perturbations promoting gut-luminal pathogens to bloom.

While we did not observe microbiota composition shifts in the LPS-exposed mice used here, oxygen species exposure induced composition shifts in some human microbiota consortia in the bioreactors. The compositional changes in the bioreactors were driven by facultative anaerobic opportunistic pathogens, like *E. coli*, able to cope with/exploit increased oxygen species abundance. For example, *E. coli* has previously been shown to utilise oxygen derived from detoxified H₂O₂ for aerobic respiration and growth[79]. In contrast, facultative

opportunistic pathogens are not present in the microbiota of our mice and can therefore not contribute to microbiota composition shifts. Importantly, bioreactor microbiota composition returned to baseline at 72 h post oxidative stress, highlighting the transient inhibitory effect of oxygen species on the gut microbiota. In LPS-exposed mice, a similar transient effect of oxidative stress on microbiota function was observed, with reduced fermentation and an increase in protein repair mechanisms. This suggests that colonisation resistance can be impacted not only by compositional alterations but also by transient functional changes in the microbiota.

The H₂O₂ concentrations used in this study have important implications for relevant physiological conditions. While H₂O₂ concentrations measured in the gut are typically in the μM range, transient spikes in concentration, potentially reaching mM concentrations near the intestinal tissue, may occur during inflammatory responses. In our experiments, μM concentrations of H₂O₂ did not significantly inhibit the growth of strains, suggesting that these levels may not be sufficient to induce detectable inhibitory effects in vitro. This could be due to the rapid reaction of H₂O₂ with anaerobic media components as well as the microbiota's inherent buffering capacity, including oxygen species scavenging and repair mechanisms that enable survival upon exposure to oxidative stress[73]. This buffering capacity might also affect gut-luminal oxygen species measurements, and therefore average μM measurements from bulk samples might not be reflective of transient oxygen species bursts. Consequently, it was necessary to add higher H₂O₂ concentrations to uncover the differential effects of oxygen species on growth and fermentation between the microbiota and opportunistic pathogens.

While we show that exposure to the ROS $H_2O_2$ is sufficient to enable pathogens to bloom, the precise oxygen species involved, and their sources remain elusive. The host relies on the production of ROS to maintain host-microbiota homeostasis and to protect against pathogen infections[42,74,80–83]. The epithelial barrier releases low levels of ROS to control commensal bacteria near the mucosal surface during homeostasis, and infection prompts the recruitment of ROS-producing phagocytes to limit systemic dissemination of pathogens[42,74,80–83]. Notably, even in the absence of key ROS generating enzymes such as CYBB and NOX1, *S.* Tm bloomed in the intestine, likely due to redundancies in ROS generation pathways. Metabolic shifts in the intestinal mucosa and the subsequent diffusion of $O_2$ from the vascular system into the gut lumen may also contribute[65]. We further demonstrate that the release of oxygen species into the gut lumen is driven by TLR4 activation, and not by PAMPs signalling via TLR5 or TLR9. A possible explanation for this difference could be cell-type specific TLR expression and/or activation of downstream signalling pathways[84]. For example, MyD88 is a shared downstream signalling mediator of TLR4, TLR5 and TLR9, whereas TLR4 can also signal via TRIF[84]. Our study prompts further investigation into the specific oxygen species types, their origins, and the signalling pathways involved in the release of oxygen species into the gut lumen during systemic immune activation.

A notable limitation of the acute LPS model employed here is its inability to fully replicate the multifaceted clinical presentation of critically ill patients, which often encompass organ damage, tissue repair and later-stage pronounced immunosuppression, a known compensating effect upon endotoxin exposure[85]. Moreover, the difference in acute susceptibility to LPS between mice and humans, along with the absence of clinical history as observed in patients, presents a challenge in directly translating our findings to human contexts. Despite these limitations, our observations phenocopy clinical observations in patients experiencing endotoxemia which undergo gut-luminal pathogen blooms during the acute inflammatory phase[6–13], highlighting the relevance of the uncovered mechanisms for the prevention of infection in patients presenting with acute systemic immune activation.

Endotoxin-driven systemic immune activation is a common hallmark of the acute phase of critical illness and is in some cases associated with gut-luminal pathogen blooms, yet the mechanisms driving these blooms frequently remain elusive[8–11]. We provide important, mechanistic insights into how gut-luminal pathogens exploit transient changes in the intestinal microenvironment caused by acute systemic immune activation to boost gut-luminal blooms, which may increase the risk for subsequent systemic dissemination[86,87]. Potential mitigation strategies include interference with gut-luminal oxygen species or the introduction of non-pathogenic competitor strains that bloom in the gut lumen but fail to survive at systemic sites. Altogether, this work demonstrates that systemic immune activation can cause gut-luminal blooms of pathogens and opportunistic pathogens, and pinpoints targets for mitigating associated risks.

## Methods
### Bacterial strains
Bacterial strains used in this study are described in Supplementary Table 1. All *S.* Tm strains are isogenic to *S.* Tm SB300[88]. The *K. pneumoniae* strain was isolated from a stool sample of a recovering Salmonellosis patient. For animal experiments, *S.* Tm, *E. coli*, and *K. pneumoniae* were routinely grown aerobically at 37 °C with agitation in lysogeny broth (LB) supplemented with the appropriate antibiotics 50 µg ml⁻¹ streptomycin (AppliChem; A1852), 15 µg ml⁻¹ chloramphenicol (AppliChem; A1806), 50 µg ml⁻¹ kanamycin (AppliChem; A1493) or 100 µg ml⁻¹ ampicillin (AppliChem; A0839). *E. faecium* was grown aerobically at 37 °C with agitation in brain heart infusion broth (BHI; Thermo Fisher Scientific; CM1135B). For hydrogen peroxide

sensitivity experiments, strains were grown in an anaerobic chamber (Coy Laboratory Products) (2% $H_2$, 12% $CO_2$, 86% $N_2$) at 37 °C in modified Gifu anaerobic medium (mGAM) broth (HyServe, 1005433-001). Strains were stored at -80 °C in peptone glycerol broth.

### Animals
All animal experiments were performed in accordance with legal and ethical regulations. Experiments were approved by Kantonales Veterinäramt Zürich under licenses ZH158/2019 and ZH108/2022. Specific pathogen–free (SPF) mice were bred under full barrier conditions in individually ventilated cages at the ETH Phenomics Centre (EPIC) of ETH Zürich. Germ-free (GF) and Oligo-MM[12] mice were bred in flexible film isolators at the isolator facility at EPIC. The following mouse lines were used: C57BL/6 J (WT; Ly5.2, in-house breeding), $Tlr4^{-/-}$ (B6.129-$Tlr4^{tm1Aki/Aki}$)[89], $Casp11^{-/-}$ (B6.B6-$Casp11^{tm1}$)[90], $Rag2^{-/-}$ $Il2rg^{-/-}$ (B6.B6($Rag2^{tm1Fwa}$)($Il2rg^{tm1Cgn}$))[91,92], $Tnf^{-/-}$ (B6.129-$Tnf^{atm1Ljo}$)[93], $Il22^{-/-}$ (B6.129S5-$Il22^{tm1Lex}$)[94], $Nos2^{-/-}$ (B6.129P2-$Nos2^{tm1Lau}$)[95], $Cybb^{-/-}$ (B6.129S-$Cybb^{tm1Din/J}$)[96] and $Nox1^{-/-}$ (B6.129X1-$Nox1^{tm1Kkr/J}$)[97]. All mice had a C57BL/6 background. All mice were maintained on the normal mouse chow (Kliba Nafag; 3430). 8- to 12-week-old mice of both sexes were randomly assigned to experimental groups to ensure generalisability of results. Cohoused littermates were used as controls where applicable.

### Animal experiments
Mice were i.v. injected with 5 µg ultrapure *S.* Tm LPS (kind gift of Otto Holst, Research Center Borstel, Germany), 5 µg ultrapure LPS from *E. coli* O111:B4 (InvivoGen; tlrl-3pelps), 12.5 µg ultrapure flagellin from *S.* Tm (InvivoGen; tlrl-epstfla-5), or 1.8 µg ODN 2395 (InvivoGen; tlrl-2395) in 100 µl PBS, unless stated otherwise. For infection experiments, bacterial strains were grown aerobically overnight at 37 °C with agitation in LB or BHI with the appropriate antibiotics. Cultures were diluted 1:20 and sub-cultured in the LB or BHI without antibiotics for 4 h at 37 °C with agitation. Bacterial cells were washed and resuspended in PBS. Mice were i.v. injected with PBS, LPS, flagellin or CpG and immediately infected with $5 \times 10^7$ c.f.u. bacterial suspension by intragastrical gavage, unless stated otherwise. Mice were euthanised by $CO_2$ asphyxiation at the indicated time points. Fresh faecal pellets, caecum content, mesenteric lymph nodes and spleens were harvested and homogenised by bead-beating for 2.5 min (min) at 25 Hz in 1 ml PBS, supplemented with 0.5% v/v Tergitol (Sigma-Aldrich; 86454) and 0.5% w/v BSA (Chemie Brunschwig; P06-1391100) for systemic organs, using the TissueLyser II (Qiagen). Bacterial loads were determined by plating on MacConkey agar (Thermo Fisher Scientific; CM0007B) or *Enterococcus* selective agar (Thermo Fisher Scientific; 11703493) containing the appropriate antibiotic.

For streptomycin pre-treatment, mice were treated with 25 mg streptomycin (AppliChem; A1852) by intragastrical gavage 24 h before infection. For neutrophil depletion, mice were i.p. injected daily with 500 µg anti-Ly6G antibody (InVivoMAb; BE0075 (Clone 1A8)) as of 1 d before infection. For macrophage depletion, mice were i.p. injected with 1 mg anti-CSF1R antibody (InVivoMAb; BE0213 (Clone AFS98)) or the respective isotype control (InVivoMAb; BE0088 (Clone HRPN)) at 4 d before infection, and subsequently 0.3 mg i.p. daily. For IFNAR blocking, mice were i.p. injected with 1 mg of anti-IFNAR antibody (InVivoMAb; BE0241 (Clone MAR1-5A3)) on the day of infection.

### PolyFermS
Two independent PolyFermS experiments were conducted under the proximal colon conditions of a healthy adult and a healthy toddler, using the Multifors 2 system (Infors AG) or DASbox system (Vaudaux-Eppendorf AG)[77,98]. The Ethics Committee of ETH Zürich exempted this study from review because the sample collection procedure was not performed under conditions of intervention and samples were anonymised. Each experimental set-up included an inoculum reactor (IR), containing immobilised human faecal microbiota gel beads at a

concentration of 30% v/v, and consecutive TRs, which were continuously inoculated with 5% v/v of the fermented effluent from the IR to establish an identical microbial profile in the TRs[99]. The IR of PolyFermS1 contained immobilised faecal microbiota of an adult donor and inoculated TR1-4 while the IR of PolyFermS2 harboured immobilised faecal microbiota from a toddler and inoculated TR5 and TR6. After a 3-day-period of continuous fermentation, the TRs were disconnected from the IR and further stabilised for another 4 days to reach metabolic stability before treatment initiation, which is defined as a variation in daily metabolite concentrations <10%. All TRs were continuously fed with a complex culture medium simulating the ileal chyme entering the human colon, that differed in bile salts and vitamins concentrations between the adult and toddler PolyFermS 1 and 2[76,99]. With a TR volume of 200 ml in the adult model and 160 ml in the toddler model, the flow rate was set to 25 and 20 ml/h, respectively to obtain a mean retention time of 8 h. The pH was controlled at 5.8 by addition of 2.5 M NaOH. All TRs were operated at 37 °C and stirring at 120 rotations per minute. The TR headspace was continuously flushed with filter-sterile $CO_2$ to ensure anaerobiosis.

TRs were inoculated with *S*. Tm at -12 h to reach a density of $1 \times 10^8$ c.f.u. $ml^{-1}$. 12 h post *S*. Tm inoculation (t0), TRs were exposed to varying hydrogen peroxide (Sigma-Aldrich; H1009) concentrations to increase the redox potential, a common measurement of the degree of anaerobiosis. TR1 in PolyFermS1 and TR5 in PolyFermS2 served as controls and did not undergo hydrogen peroxide treatment, the redox potential in these control TRs was -400 mV. TR content in TR2 was exposed to a final hydrogen peroxide concentration of 1.5 mM increasing the redox potential to -300 mV, TR3 3 mM hydrogen peroxide to redox potential of 0 mV, TR4 8 mM hydrogen peroxide to redox potential of +200 mV (TR4), and TR6 12.5 mM hydrogen peroxide to redox potential of +270 mV. Sampling of TR effluent was performed every 3 to 12 h, for 16S rRNA gene analysis, metabolite analysis and *S*. Tm density analysis. *S*. Tm densities were determined by plating on MacConkey agar containing the appropriate antibiotic.

### Isolation and whole genome sequencing of the *K. pneumoniae* clinical isolate (T821)

Aliquots of a frozen stool sample from a recovering Salmonellosis patient were homogenised in PBS, diluted, and plated on MacConkey agar (Thermo Fisher Scientific; CM0007B). Morphologically distinct colonies were picked and grown in LB liquid culture. Genomic DNA was isolated from this enrichment culture using the DNeasy Blood & Tissue Kit (QIAGEN; 69504). Library preparation and sequencing were performed by Novogene on the Illumina NovaSeq 6000 platform. Reads were assembled to contigs using the CLC Genomics Workbench 20.0.4 software (QIAGEN). Species were identified by multi-locus sequence typing[100].

### Quantification of *S*. Tm growth rates in the caecum

*S*. Tm growth rates in the caecum were assessed using replication-incompetent plasmid pAM34[101]. pAM34 is a ColE1-like vector in which the replication of the plasmid is under the control of the LacI repressor, therefore plasmid replication only occurs in the presence of isopropyl β-D-1-thiogalactopyranoside (IPTG)[101]. *S*. Tm pAM34 was cultured for 12 h in LB medium supplemented with 1 mM IPTG (Biosynth; I-8000). Cultures were diluted 1:20 and sub-cultured in LB without IPTG for 3 h at 37 °C shaking. Mice were orally pre-treated with streptomycin 1 d before infection or i.v. injected with PBS or LPS and subsequently infected with $5 \times 10^7$ c.f.u. *S*. Tm pAM34. To generate a standard curve relating plasmid loss to the number of generations, the inoculum was serially diluted in LB and cultured for 12 h at 37 °C shaking. The fraction of pAM34 harbouring bacteria in the caecal content and standard curve cultures was determined by selective plating on MacConkey agar plates (Thermo Fisher Scientific; CM0007B) containing 100 μg $ml^{-1}$ ampicillin (AppliChem; A0839) and

1 mM IPTG and on MacConkey agar plates containing 50 μg $ml^{-1}$ streptomycin (AppliChem; A1852). The number of *S*. Tm generations in the caecal content was estimated by interpolation from the matched standard curve.

### Bacterial flow cytometry

Activity of the *sicA* promoter was analysed in mice using a plasmid-based *PsicA-gfp* reporter[102]. Fresh caecum content was collected 24 h.p.i. and diluted in PBS. Caecum content was incubated for 1 h at room temperature with 2 μg $ml^{-1}$ of chloramphenicol (AppliChem; A1806) to inhibit protein synthesis and allow GFP to fully mature. *S*. Tm was stained with a human anti-*S*. Tm O12 antibody (hSTA5; kind gift of Antonio Lanzavecchia, Institute for Research in Biomedicine, Bellinzona, Switzerland), and goat anti-human IgG AF647 antibody (Jackson ImmunoResearch Europe; 109-605-098). Fluorescence was measured with a Cytoflex flow cytometer (Beckman Coulter) acquired with CytExpert software v.2.5. Gating strategy provided in Supplementary Fig. 7a.

### Lipocalin-2 ELISA

Caecal contents were homogenised by bead-beating for 2.5 min at 25 Hz in 500 μl PBS using the TissueLyser II (Qiagen). The homogenate was then centrifuged at 16'000 x *g* for 5 min. Lipocalin-2 was measured in the supernatant by ELISA DuoSet Lipocalin ELISA kit (R&D Systems; DY1857) according to the manufacturer's instructions. Briefly, capture antibodies were coated onto a 96-well Nunc Maxisorb plate (Sigma-Aldrich; M9410), followed by incubation with supernatants, detection antibodies, and substrate solution. The absorbance was measured at 405 nm using the Spectramax Plus (Molecular Devices), and Lipocalin-2 concentrations were calculated based on a standard curve.

### Histopathology

Caecal tissues were embedded in Tissue-Tek OCT medium (Sysmex; TT 4583) and were snap-frozen in liquid nitrogen and stored at -80 °C until sectioning. Cryosections of 10 μm thickness were prepared using a cryostat NX50 (Thermo Fisher Scientific) and mounted onto glass slides for haematoxylin and eosin (H&E) staining. For staining, sections were fixed in Wollman's fixative (95% v/v ethanol, 5% v/v acetic acid) for 30 s, followed by sequential washing in tap water and deionized water. Nuclei were stained with Mayer's haematoxylin (Merck; 109249) for 15 min, then blued in tap water. Sections were briefly destained in 70% v/v ethanol with 1% v/v hydrogen chloride in water, followed by bluing in tap water and deionized water, and dehydration in 70% v/v ethanol in water, and 90% v/v ethanol in water. Cytoplasmic staining was performed with alcoholic Eosin containing Phloxin (Sigma-Aldrich; HT1103128) for 3 min, followed by sequential dehydration in 90% v/v ethanol in water, 100% v/v ethanol in water, and xylene. Finally, slides were mounted with Entellan (Merck, 1.07961.0100) for long-term preservation and microscopic evaluation.

### Bulk RNAseq

Caecum tissue was snap-frozen in RNAlater (Sigma-Aldrich; R0901-500ML). RNA isolation was performed with the RNeasy Mini Kit (Qiagen; 74104) according to the manufacturer's instructions, including DNAse digestion. mRNA sequencing was performed by the Functional Genomics Center Zürich (FGCZ; ETH Zürich) on the Illumina NovaSeq 6000 platform. Reads were mapped to the mouse genome using STAR[103]. Reads were quantified using featureCounts[104]. Differential expression analysis was performed using edgeR[105]. Over-representation analysis was performed using WebGestalt 2019[106]. Data was visualised using ggplot2.

### Quantification of oxygen species in caecal contents

Freshly isolated caecal content samples were diluted in deionized water and vortexed extensively. The caecal suspension was

centrifuged at 1800 x *g* for 5 min. Oxygen species concentrations were measured in the supernatants with the Amplex Red/Horseradish Peroxidase assay kit (Thermo Fisher Scientific; A22188) according to the manufacturer's instructions. The absorbance was measured at 560 nm using the Spectramax Plus (Molecular Devices), and oxygen species concentrations were calculated based on a standard curve.

## Quantification of WITS-barcoded mutant strains

Mice were infected with a pool of WITS-barcoded wild-type and mutant *S*. Tm strains mixed in an equal ratio[107]. 50% of the *S*. Tm pool contained WITS-barcoded wild-type strains to ensure comparable disease kinetic in separate experiments. Caecal content was collected at 24 h.p.i. and was enriched for *S*. Tm in LB supplemented with 50 µg ml$^{-1}$ streptomycin at 37 °C with agitation for 4 h. Cells were pelleted and stored at -20 °C. gDNA was extracted from the pellets using the QIAamp DNA Mini kit (Qiagen; QIAamp DNA Mini) according to the manufacturer's instructions. RT-qPCR was performed using FastStart Universal SYBR Green Master (Rox) reagents (Roche; 4913914001) and WITS-barcode-specific primers at a final concentration of 1 µM. RT-qPCR was performed on a StepOne Plus Cycler (Thermo Fisher Scientific)[107]. WITS-barcode ratios were calculated using a standard curve generated using gDNA extracted from a single barcoded strain. The population size of each tagged strain was calculated by multiplying the bacterial density in c.f.u. g$^{-1}$ with the WITS-barcode ratio. The CI was calculated by dividing the population size of the mutant by the population size of the wild-type strain, normalised to the mutant-to-WT ratio of the inoculum.

## Quantification of caecal content microbiota densities

The whole intestinal tract was harvested and transferred into an anaerobic chamber (Coy Laboratory Products). Caecal contents were serially diluted in BHI broth (Thermo Fisher Scientific; CM1135B). Dilutions were plated on BHI plates supplemented with 5 µg ml$^{-1}$ hemin (Sigma-Aldrich; H9039-1G), 1 µg ml$^{-1}$ menadione (Sigma-Aldrich; M5625-25G), 0.5 mg ml$^{-1}$ L-cysteine hydrochloride (Sigma-Aldrich; C1276-10G) and 10% v/v sheep blood (Thermo Fisher Scientific; sheep blood). Plates were incubated anaerobically at 37 °C for 48 h.

## 16S rRNA gene sequencing of caecal content

Caecal content collected from mice was immediately snap-frozen and stored at -20 °C. For gDNA extraction, caecal content was homogenised in RLT buffer with a 3 mm metal bead using a TissueLyser II (Qiagen) for 3 min at 30 Hz. Next, 0.1 mm zirconia beads were added, and cells were disrupted using a TissueLyser II (Qiagen) for 3 min at 30 Hz, two times. The supernatant was transferred onto a DNA column from the AllPrep DNA/RNA Mini Kit (Qiagen; 80204) and gDNA was extracted according to the manufacturer's instructions. The V4 region of the 16S rRNA gene was amplified with primers 806 R (5'-GGAC-TACNVGGGTWTCTAAT-3') and 515 F (5'-GTGYCAGCMGCCGCGGTAA-3')[108,109]. Illumina Unique Dual Indexing Primers (UDP) were used for library multiplexing. Library sequencing was performed using an Illumina NextSeq2000 flow cell with a P1 reagent kit for 2 x 300 bp PE-reads with a target fragment size of 450 bp. Sequencing was performed at the FGCZ. Reads were trimmed using cutadapt and further analysed using the dada2 pipeline[110,111]. The remaining ASVs were taxonomically annotated using IDTAXA in combination with the Silva v138 database[112,113].

## Microbiota transcriptome analysis

Caecal content collected from Oligo-MM[12] mice at 6 h post-PBS or LPS exposure was immediately flash frozen and stored at -80 °C. RNA was isolated using the RNeasy PowerMicrobiome Kit (Qiagen; 26000-50), according to the manufacturer's instructions. Bacterial ribosomal RNA removal, library preparation and sequencing (pair-end) on the Illumina NovaSeq PE150 platform were performed by Biomarker Technologies

(Münster, Germany). The resulting raw reads were cleaned by removing adaptor sequences, low-quality-end trimming and removal of low-quality reads using BBTools v 38.18 (Bushnell, B. BBMap. Available from: https://sourceforge.net/projects/bbmap/). The exact commands used for quality control can be found on the Methods in Microbiomics webpage (Sunagawa, S. Data Preprocessing - Methods in Microbiomics 0.0.1 documentation. Available from: https://doi.org/10.5281/zenodo.15019381). Reads were mapped to the concatenated genomes Oligo-MM[12] bacterial strains with bowtie2[114] and quantified using featureCounts[104]. Differential expression analysis was performed using DESeq2[115] using taxon-specific normalisation[116].

## Quantification of amino acids in caecal contents and reactor effluent

For amino acid quantification in caecal contents, freshly isolated caecal content samples were normalised to 25 mg ml$^{-1}$ in PBS, were homogenised using a TissueLyser II (Qiagen) at 25 Hz for 2 min, and were centrifuged for 5 min at 18,000 x *g*. For amino acid quantification in TR effluent, 1 ml TR effluent was collected and centrifuged for 5 min at 18,000 x *g*. Caecal content and TR effluent supernatants were cooled and were centrifuged at 18,000 x *g* for 30 min to obtain a particle free supernatant. U-$^{13}$C, $^{15}$N L-glutamate (Cambridge Isotope Laboratories; CNLM-554-H-0.5) was added to supernatants as internal standard. Samples were diluted in hydrophilic interaction liquid chromatography (HILIC) buffer and were stored at -80 °C until measurement.

LC-MS was performed on an Ultimate 3000 ULPC instrument (Thermo Fisher Scientific) hyphenated to a Qexative plus mass spectrometer (Thermo Fisher Scientific). HILIC-based separation was performed on an AQUITY BEH NH2 column (particle size 1.7 µm; 100 x 2.1 mm, Waters) with 10 mM ammonium formate (Sigma-Aldrich; 70221-25G-F) in water:acetonitrile (Sigma-Aldrich; 02000076) (50:50) (solvent A) and 10 mM ammonium formate in acetonitrile:water:methanol (Sigma-Aldrich; A456-212) (95:5:5) as mobile phase. Formic acid (Thermo Fisher Scientific; 10596814) was added to both mobile phases to shift pH. The eluent solvent B was used for the following multi-step gradient: 0 min: 84.3 %; 1.5 min: 84.3 %; 5.5 min: 5.3 %; 7.5 min: 5.3 %; 8 min: 84.3%, 10 min: 84.3%. Volumes of 2 µL were injected into the HPLC with a flow rate of 500 µl min$^{-1}$. Mass analysis was performed in the positive FTMS mode at a mass resolution of 70,000 (at m/z 200) applying heated electro-spray ionisation (HESI) at 3.5 kV. Other source settings: S-lens RF level, 50; sheath gas, 50; aux gas, 20; sweep gas, 0; aux gas heater, 350 °C. To determine relative amino acid levels, all extracted ion chromatogram (EIC) peaks were normalised to internal standard U-$^{13}$C, $^{15}$N L-glutamate peak.

## Quantification of SCFAs in caecal contents

For SCFA quantification, freshly isolated caecal content samples were homogenised in 70% v/v isopropanol in HPLC-grade water by bead-beating using a TissueLyser II (Qiagen) at 25 Hz for 1 min and were normalised to 25 mg ml$^{-1}$ in 70% v/v isopropanol in HPLC-grade water. Acetate D3 (Eurisotop; DLM-3126-25), propionate D5 (Eurisotop; DLM-1919-5) and butyrate D7 (Eurisotop; DLM-1508-5) internal standard mix was added to caecal content supernatants. The samples were derivatised with 5 mM 3-Nitrophenylhydrazine hydrochloride (Sigma-Aldrich; N21804-5G) and 3 mM N-(3-dimethylaminopropyl)-N'-ethyl-carbodiimide hydrochloride (Sigma-Aldrich; 03449-5 G) at 40 °C for 30 min. The reaction was quenched with formic acid (Thermo Fisher Scientific; A117-50) to a final concentration of 0.02% v/v. Samples were diluted in 70% v/v isopropanol in HPLC-grade water and were stored at -80 °C until measurement.

LC-MS was performed on an Ultimate 3000 ULPC instrument (Thermo Fisher Scientific) hyphenated to a QExactive plus mass spectrometer (Thermo Fisher Scientific). LC separation was performed on a Kinetex XB C18 column (particle size 1.7 µm; 50 x 2.1 mm, Phenomenex) with 0.1% v/v formic acid (Thermo Fisher Scientific; A117-50)

in HPLC-grade water (solvent A) and 0.1% v/v formic acid in acetonitrile (Sigma-Aldrich; 02000076) as mobile phase. The eluent solvent B was used for the following multi-step gradient: 0 min: 2%; 3 min: 95%; 5 min: 95%; 5.3 min: 2%. Subsequently, the column was equilibrated for 2 min at the initial condition. Volumes of 2 μL were injected into the HPLC with a flow rate of 500 μl min⁻¹. Mass analysis was performed in the negative FTMS mode at a mass resolution of 70,000 (at m/z 200) applying HESI with a spray voltage of − 2.7 kV. Other source settings: S-lens RF level, 50; sheath gas, 50; aux gas, 20; sweep gas, 0; aux gas heater, 350 °C. Absolute quantification of SCFAs was performed based on standard curves using the peak area ratio of unlabelled and corresponding labelled internal chemical standard with the following isotopic labelling: acetate D3, propionate D5 and butyrate D7.

### Quantification of hydrogen levels and food consumption

The isolator-housed TSE PhenoMaster system was used to measure ambient hydrogen levels and food consumption, according to manufacturer instructions[72]. Mice were imported into TSE PhenoMaster metabolic cages and were single-housed and adapted to the cages. Calorimetry was recorded every 24 min for individual cages for three light-dark cycles. Mice were injected with PBS or LPS at 32 h.

### Determination of bacterial strain hydrogen peroxide sensitivity

Oxygen species sensitivity testing of intestinal microbes was performed using hydrogen peroxide (Merk Millipore; 88597) as ROS inducing agent. In brief, all experiments were performed under anoxic conditions with pre-reduced mGAM (HyServe; 1005433-001) at 37 °C in an anaerobic chamber (Coy Laboratory Products Inc, 2% H₂, 12% CO₂, rest N₂). For IC25/75 determination, hydrogen peroxide treatment plates were prepared freshly in U-bottom plates (Thermo Fisher Scientific; 168136) with pre-reduced hydrogen peroxide aliquots in 2-fold dilution steps, covering a concentration range between 12 and 0.0375 mM. All plates contained untreated control wells. Strains were cultured twice overnight before adjusting to the starting OD of 0.05 at 578 nm. After inoculation, the plates were quickly sealed with breathable membranes (Breathe-Easy, Sigma-Aldrich; Z380059) and incubated without shaking. Growth curves were tracked by measuring the OD578 every hour for 18 h after 30 s of linear shaking with a microplate spectrophotometer (EON, Biotek) coupled to a microplate stacker (Biostack 4, Biotek). All screening experiments were performed in 2 biological and 2 technical replicates. The analysis of the growth curves was performed with the R package 'neckaR' (https://github.com/Lisa-Maier-Lab/neckaR)[117,118]. All growth curves were truncated at the transition from exponential to stationary phase of untreated control samples. We performed growth curve quality control (for example, removing time points with spikes in the OD) and the area under the curve (AUC) was calculated for all curves using the trapezoidal rule. For baseline correction, we assumed a constant shift which was then subtracted from all time points. The AUCs were normalised to the untreated controls by dividing the treated AUC to the mean AUC of untreated controls within the same plate. Using the median AUC over all replicates of individual concentrations, the lowest hydrogen peroxide concentrations inhibiting at least 25% (IC25) and 75% (IC75) bacterial growth were determined. To compare whether intestinal pathogens were more resistant than commensal microbiota members, we performed a two-proportions z-test using R.

### Quantification of metabolites in reactor effluent

1 ml TR effluent was collected and was centrifuged at 18,000 x $g$ for 10 min at 4 °C. Supernatants were stored at -20 °C until analysis. Before HPLC analysis, the supernatants were filtered through a nylon membrane (0.2 μm). HPLC was performed on Accela (Thermo Fisher Scientific) equipped with a Carbon-H cartridge (4 x 3.0 mm) connected to a Resex ROA-Organic Acid column (300 x 7.8 mm) (Phenomenex Helvetia GmbH) and an Accela refraction index detector (Thermo Fisher Scientific). Volumes of 20 μl were injected into the HPLC with a flow rate of 0.4 ml min⁻¹ at a constant column temperature of 40 °C using 10 mM $H_2SO_4$ as an eluent. Metabolites were quantified using external standards by comparing the retention time.

### 16S rRNA gene sequencing of reactor effluent

1 ml TR effluent was collected, and bacteria were pelleted at 18'000 x $g$ for 10 min at 4 °C. Bacterial pellets were stored at -20 °C until further processing. gDNA was extracted with the FastDNA Spin Kit for Soil (MP Biomedicals; 116560200-CF) according to the manufacturer's instructions. The V4 region of the 16S rRNA gene was amplified and barcoded with primers 806 R (5′-GGACTACHVGGGTWTCTAAT-3′) and 515 F (5′-GTGCCAGCMGCCGCGGTAA-3′)[119]. Sequencing was performed using an Illumina MiSeq flow cell with a V2 reagent kit (Illumina, MS-102) for 2 × 250 bp paired-end Nextera chemistry supplemented with 10% PhiX at the Genetic Diversity Center (ETH Zürich, Switzerland). The raw fastq files from the two separate sequencing runs were processed together into one single phyloseq object with the metabaRpipe (v0.9) pipeline using dada2[111]. The taxonomy was assigned according to the SILVA v138.1 database[113]. Data was visualised using ggplot2.

### 16S rRNA copy number quantification of reactor effluent

gDNA isolated for 16S rRNA gene sequencing was used for 16S rRNA copy number quantification. RT-qPCR was performed using SensiFast SYBR No Rox Mix and 16S-specific primers at a final concentration of 500 nM[120]. RT-qPCR was performed on a LightCycler 480 (Roche). 16S rRNA copy number was calculated using a standard curve generated using linearised plasmids containing the full 16S rRNA gene of *Faecalibacterium prausnitzii*.

### RT-qPCR

Caecum tissue was snap-frozen in RNAlater (Sigma-Aldrich; R0901-500ML). RNA isolation was performed with the RNeasy Mini Kit (Qiagen; 74104) according to the manufacturer's instructions, including DNAse digestion. cDNA was synthesised with the RT² HT First Strand kit (Qiagen; 330411). RT-qPCR was performed using FastStart Universal SYBR Green Master (Rox) reagents (Roche; 4913914001) and RT² qPCR primers (Qiagen) (Supplementary Table 2), on a StepOne Plus Cycler (Thermo Fisher Scientific). mRNA levels were normalised to *Actb* and calculated with the $2^{-\Delta CT}$-method.

### Lamina propria flow cytometry

Caecum tissue was harvested and washed in PBS. Epithelial cells were dislodged in PBS supplemented with 5 mM EDTA (Thermo Fisher Scientific; AM9261), 15 mM HEPES (Thermo Fisher Scientific; 15630080) and 10% v/v heat-inactivated FCS (Thermo Fisher Scientific; 10500064). Tissue was digested into a single-cell suspension in RPMI (Thermo Fisher Scientific; 1-41F01-I) supplemented with 1 mg collagenase VIII (Sigma-Aldrich; C2139-1G) and 0.2 mg DNAse I (Sigma-Aldrich; 11284932001). Cells were incubated in 1 μg per sample Mouse BD Fc Block (BD Biosciences; 553142) in 10% v/v Brilliant stain buffer (BD Biosciences; 566349)/FACS buffer for 5 min at 4 °C. Followed by staining with the antibody mix for 30 min at 4 °C (Supplementary Table 3). Samples were measured on a LSR Fortessa (BD Biosciences), and data was analysed with FlowJo V10 (TreeStar). Gating strategy provided in Supplementary Fig. 7b.

### Statistics and reproducibility

A minimum of five mice per group was used in this study. Sample sizes were chosen based on institutional guidelines and in adherence to the 3Rs principles (Replacement, Reduction, and Refinement) to uphold ethical standards in animal research. No statistical method was used to predetermine the sample size. A minimum of five mice per group was deemed sufficient to observe consistent and reproducible results across experimental groups, based on prior studies with similar

experimental designs. No data were excluded from the analyses. Mice were randomly allocated to different treatment groups. Blinding was not applicable, as investigators needed to identify the cages of mice for subsequent treatments or infections with respective bacterial strains. All experiments were performed at least twice, with all attempts at data replication being successful.

Data was plotted using GraphPad Prism v9.0 for Windows (GraphPad Software, La Jolla California USA, www.graphpad.com) or ggplot2 in R v4.3.2 with R Studio v2023.09.1. Statistical testing was performed using the two-sided Mann-Whitney U test, two-sided Kruskal-Wallis test with Dunn's multiple test correction, or two-sided Wald test with Benjamini-Hochberg multiple test correction, as appropriate. $P$ values of <0.05 were considered as statistically significant ($^*P < 0.05$, $^{**}P < 0.01$, $^{***}P < 0.001$, $^{****}P < 0.0001$).

### Reporting summary
Further information on research design is available in the Nature Portfolio Reporting Summary linked to this article.

## Data availability
The raw sequencing data generated in this study have been deposited in the European Nucleotide Archive (ENA) under accession number PRJEB82444 and in the NCBI BioSample database under accession number SAMN45075194. Source data are provided with this paper.

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

## Acknowledgements

We thank Annelies Zinkernagel and Simone Becattini for the helpful scientific discussions. We thank all members of the Hardt, Sunagawa and Slack labs for helpful comments and discussions. We acknowledge the staff at the ETH animal facilities (EPIC and RCHCI; especially Manuela Graf, Katharina Holzinger, Dennis Mollenhauer, Sven Nowok & Dominik Bacovcin) for outstanding support of our animal work. We thank Elisa Cappio Barazzone and Giorgia Greter for technical assistance in conducting the Phenomaster experiments. We thank for Alfonso Die for the technical assistance for the PolyFermS metabolite measurements. We thank Abigail Dustour and Nabila Bittar for technical assistance in conducting the mouse experiments. We are grateful to Otto Holst for providing ultrapure LPS. This work has been funded by grants from the Swiss National Science Foundation (310030_192567, 10.001.558 to WDH), NCCR Microbiomes (51NF40_180575 to ES, JAV, SS and WDH) and the European Union's Horizon 2020 research and innovation programme (Marie Skłodowska-Curie No 956279 to WDH). AH was supported by an EMBO Postdoctoral Fellowship (ALTF 179-2021) and a grant by the European Crohn´s and Colitis Organization (ECCO) (PROP–1495). L.Bo. and L. M. were support by the German Research Foundation (EXC2124). ES was supported by Swiss National Science Foundation (40B2-0_180953, 310030_185128) and Basel Research Centre for Child Health Multi-Investigator Project 2020 (BRCCH_MIP: Microbiota Engineering for Child Health).

## Author contributions

S.K., L.Bi, E.S., S.S., J.A.V., L.M., C.L., A.H. and W.-D.H. conceived and designed the experiments. S.K., D.M., L.W., L.Bi., L.Bo., P.C., P.K., A.S., B.D.N., M.B., Y.S., M.C., E.G. and A.H. performed the experiments and analysed the data. M.K.-M.H., C.C. and B.G. provided the *Klebsiella pneumoniae* T821 strain. S.K., A.H. and W.-D.H. wrote the manuscript. All authors read, commented, and approved the manuscript.

## Competing interests

The authors declare no competing interests.
