## [Transparent Peer Review file · Nature Communications]

Sublethal systemic LPS in mice enables gut-luminal pathogens to bloom through oxygen species-mediated microbiota inhibition

Corresponding Author: Professor Wolf-Dietrich Hardt

Version 0:

Reviewer comments:

Reviewer #1

(Remarks to the Author)

In this manuscript, Kroon et al investigate mechanisms of gut microbiota changes during endotoxemia in a mouse model. LPS is administered at low, sublethal concentrations by iv, and subsequent changes to the gut microbiota and colonization with experimentally introduced gut bacteria is recorded. Curiously, prior endotoxemia decreases colonization resistance to *S. Tm* and opportunistic pathogens such as *E. coli* and *Enterobacter*. LPS-induced decrease in colonization resistance is dependent on TLR4, and cannot be achieved with other TLR ligands. Systemic LPS administration results in the induction of host responses in the gut, such as cytokine secretion and release of reactive oxygen species, although no overt signs of inflammation are noted at these time points. *Salmonella* uses respiration to colonize the gut of endotoxemic mice. No over change in the gut microbiota occur upon systemic LPS administration, however, metabolic output, in particular fermentative processes, of the microbiota is altered. The effect of reactive oxygen species on human gut microbiota is analyzed in a fermenter model; consistent with the mouse model, exposure to H₂O₂ decreased metabolic output, and increased *Salmonella* populations. The authors conclude that transient endotoxemia results in a short pulse of oxidative stress in the gut, which in turn decreases colonization resistance by pathogenic *Enterobacteriaceae*.

This study provides novel, mechanistic insights into how endotoxemia influences host-microbe interactions in the intestinal tract, and provides a potential explanation as to why patients with acute critical illness exhibit changes in their gut microbiome. As such, these original findings should appeal to a broad audience, including clinicians, infectious diseases researchers, and immunologists. The overall conclusions are well supported by the data, and the study design is elegant. There is a clear rationale to focus on both professional and opportunistic pathogens. I only have minor suggestions for improvements, see below.

1. This study does a great job delineating why endotoxemia leads to changes in host-microbe interactions in the gut, but the consequences of this altered interaction are not very clear. For example, in the experiments shown in Fig. 1F - H, did the authors investigate as to whether increased gut colonization by opportunistic microbes had any functional consequences, e.g. increased dissemination to systemic sites? If this data was available (I may have missed it in the manuscript), it would be important to show.

2. Chanin et al, *Cell Host Microbe*. 2020 showed that *E. coli* can utilize reactive oxygen species to support respiration, a finding that could be of relevance for the discussion.

Reviewer #2

(Remarks to the Author)

In their study, the authors used a murine model of acute endotoxin exposure to show how systemic immune activation leads to gut-luminal pathogen blooms via oxidative respiration, independently of overt intestinal inflammation and drastic microbiota shifts. These findings significantly improve our understanding of the mechanisms at play during acute LPS exposure.

Main comment:

The authors' approach of using H₂O₂ to simulate oxidative stress in their in vitro assays is a reasonable method to explore microbial responses to reactive oxygen species (ROS). However, there are concerns regarding whether the H₂O₂

concentrations used in their experiments truly mimic the levels observed in the human gut during LPS exposure. The concentrations of H₂O₂ (1.5-8 mM) used in the in vitro experiments appear quite high compared to physiological levels typically found in the human gut, even under inflammatory conditions. In vivo, the concentration of H₂O₂ and other ROS is usually in the micromolar range. The use of millimolar concentrations might not accurately reflect the actual oxidative stress environment in the gut.

Same in the experiments where bacteria were exposed to different levels of H₂O₂. The concentrations of H₂O₂ used in the study (0.3 to 3 mM) are significantly higher than those typically produced in human gut conditions, even during acute oxidative stress and inflammation. While useful for understanding the mechanisms of microbial response to oxidative stress, these levels do not mimic physiological H₂O₂ concentrations in humans.

Version 1:

Reviewer comments:

Reviewer #1

(Remarks to the Author)

The authors have adequately addressed my concerns.

Reviewer #2

(Remarks to the Author)

The authors have clearly addressed my comments, and I have no further remarks.

Point by Point Response to the Reviewer Comments:

Reviewer #1 (Remarks to the Author):

In this manuscript, Kroon et al investigate mechanisms of gut microbiota changes during endotoxemia in a mouse model. LPS is administered at low, sublethal concentrations by iv, and subsequent changes to the gut microbiota and colonization with experimentally introduced gut bacteria is recorded. Curiously, prior endotoxemia decreases colonization resistance to *S. Tm* and opportunistic pathogens such as *E coli* and Enterobacter. LPS-induced decrease in colonization resistance is dependent on TLR4, and cannot be achieved with other TLR ligands. Systemic LPS administration results in the induction of host responses in the gut, such as cytokine secretion and release of reactive oxygen species, although no overt signs of inflammation are noted at these time points. Salmonella uses respiration to colonize the gut of endotoxemic mice. No over change in the gut microbiota occur upon systemic LPS administration, however, metabolic output, in particular fermentative processes, of the microbiota is altered. The effect of reactive oxygen species on human gut microbiota is analyzed in a fermenter model; consistent with the mouse model, exposure to H₂O₂ decreased metabolic output, and increased *Salmonella* populations. The authors conclude that transient endotoxemia results in a short pulse of oxidative stress in the gut, which in turn decreases colonization resistance by pathogenic Enterobacteriaceae. This study provides novel, mechanistic insights into how endotoxemia influences host-microbe interactions in the intestinal tract, and provides a potential explanation as to why patients with acute critical illness exhibit changes in their gut microbiome. As such, these original findings should appeal to a broad audience, including clinicians, infectious diseases researchers, and immunologists. The overall conclusions are well

supported by the data, and the study design is elegant. There is a clear rationale to focus on both professional and opportunistic pathogens. I only have minor suggestions for improvements, see below.

Response: We are very pleased with this positive assessment of our work.

1. This study does a great job delineating why endotoxemia leads to changes in host-microbe interactions in the gut, but the consequences of this altered interaction are not very clear. For example, in the experiments shown in Fig. 1F - H, did the authors investigate as to whether increased gut colonization by opportunistic microbes had any functional consequences, e.g. increased dissemination to systemic sites? If this data was available (I may have missed it in the manuscript), it would be important to show.

Response: Thank you very much for this insightful comment. We agree and have collected data on the systemic dissemination of *S. Typhimurium* and *E. coli*. This data is presented in the revised Extended Data Figure 1 c-f. In case of a pathogen capable of surviving phagocyte encounters at systemic sites (like *S. Typhimurium*), we can indeed see that luminal blooms can translate into elevated systemic spread. In contrast, in case of *E. coli* strains lacking virulence factors that permit systemic survival or growth, the gut-luminal bloom does not translate into elevated systemic spread, at least not under the conditions chosen in our experiments. This data is discussed in the revised manuscript (lines 158-167)

2. Chanin et al, Cell Host Microbe. 2020 showed that *E. coli* can utilize reactive oxygen species to support respiration, a finding that could be of relevance for the discussion.

Response: We agree and are discussing this important paper in the revised discussion (lines 475-476).

Reviewer #2 (Remarks to the Author):

In their study, the authors used a murine model of acute endotoxin exposure to show how systemic immune activation leads to gut-luminal pathogen blooms via oxidative respiration, independently of overt intestinal inflammation and drastic microbiota shifts. These findings significantly improve our understanding of the mechanisms at play during acute LPS exposure.

Response: Thank you very much for this positive evaluation of our findings.

Main comment:

1. The authors' approach of using H₂O₂ to simulate oxidative stress in their in vitro assays is a reasonable method to explore microbial responses to reactive oxygen species (ROS). However, there are concerns regarding whether the H₂O₂ concentrations used in their experiments truly mimic the levels observed in the human gut during LPS exposure. The concentrations of H₂O₂ (1.5-8 mM) used in the in vitro experiments appear quite high compared to physiological levels typically found in the human gut, even under inflammatory conditions. In vivo, the concentration of H₂O₂ and other ROS is usually in the micromolar range. The use of millimolar concentrations might not accurately reflect the actual oxidative stress environment in the gut.

Same in the experiments where bacteria were exposed to different levels of H₂O₂. The concentrations of H₂O₂ used in the study (0.3 to 3 mM) are significantly higher than those typically produced in human gut conditions, even during acute oxidative stress and inflammation. While useful for understanding the mechanisms of microbial response to oxidative stress, these levels do not mimic physiological H₂O₂ concentrations in humans.

Response:

We appreciate the reviewer's insightful comments regarding the use of H₂O₂ concentrations in our in vitro assays. We understand the concern that the concentrations used (0.375-12 mM) in our in vitro assay appear higher than the physiological levels typically found in the human gut under inflammatory conditions. Our rationale for using higher concentrations is based on the bacterial strains' buffering capacity (i.e. when grown under anaerobic conditions). When H₂O₂ is introduced, most H₂O₂ will oxidize the anaerobic bacteria, leading to swift depletion of the added H₂O₂. In addition, H₂O₂ is depleted by the bacteria's antioxidant defenses and enzymatic activities, such as catalase and peroxidase. This results in a substantial reduction in the "measurable" H₂O₂ concentration, thereby mimicking the transient spikes of ROS that bacteria might encounter in vivo. To illustrate this effect, we are showing the time course of the redox potential which we had recorded in our PolyFermS experiments (Extended Data Fig. 5b,c). These redox measurements show a sharp increase of the redox potential from approx. -400 mV in steady state to -200 (or higher, if we use more H₂O₂) right after H₂O₂ injection. When we added 1.5 mM H₂O₂, the redox potential rose to -200 mV and returned to nearly the pre-injection redox within <30 min. Importantly, these low H₂O₂ concentrations are already associated with transient microbiota shifts in the PolyFermS experiments (Fig. 6e,f).

How does this translate to the H₂O₂ concentrations measured in human patients suffering from gut inflammation? When measuring the H₂O₂ concentrations in the gut, the H₂O₂ released into the gut has already been diminished by exactly those buffering reactions. This explains why the measured H₂O₂ concentrations in the patients' gut lumen are significantly lower than the 1.5 mM H₂O₂ (or more), which we have added to the PolyFermS fermenters, or the H₂O₂ used in our microbiota strain growth assays (Fig. 5). This is explained more clearly in the revised manuscript (lines 361-375; lines 387-394; lines 485-497). Based on these considerations, we expect that the real free H₂O₂ concentrations in the gut of LPS injected mice (100 micromolar oxygen species; Fig. 2f), in our fermenter assays and in the inflamed human intestine are in fact quite similar.

Overall, it is of course challenging to directly measure H₂O₂ concentrations in the human gut due to dynamic interactions between the host's ROS production and the microbial community's buffering capacity, with concentrations being modulated by microbial and host antioxidant systems. This makes it difficult to capture the precise concentrations experienced by the gut microbiota. Our experimental design aims to create a robust model to understand bacterial responses to oxidative stress by initially applying higher concentrations of H₂O₂ which rapidly drop to levels approximating concentrations measured in the human gut. This approach allows us to study microbial response mechanisms to transient oxidative stress, which is critical for understanding their survival and function in the gut during inflammatory states. We believe this methodology provides valuable insights into microbial dynamics under oxidative stress and offers a relevant framework for exploring microbial resilience and adaptation mechanisms in the gut.